

# Representation of dissolved organic carbon in the JULES land surface model (vn4.4_JULES-DOCM)

Mahdi Nakhavali[1], Pierre Friedlingstein[1], Ronny Lauerwald[1], Jing Tang[2,3], Sarah Chadburn[1,4], Marta Camino-Serrano[5], Bertrand Guenet[6], Anna Harper[1], David Walmsley[7], Matthias Peichl[8], Bert Gielen[9]

[1]University of Exeter, Exeter, EX4 4QE, United Kingdom
[2]Terrestrial Ecology Section, Department of Biology, University of Copenhagen, Copenhagen, Denmark
[3]Centre for Permafrost, University of Copenhagen, Copenhagen, Denmark
[4]University of Leeds, School of Earth and Environment, Leeds, United Kingdom
[5]CREAF, Barcelona, Catalonia
[6]IPSL-LSCE, Gif-sur-Yvette, France
[7]Leuphana University Lüneburg, Germany
[8]Swedish University of Agricultural Sciences, Department of Forest Ecology and management, Umeå, Sweden
[9]University of Antwerp, Antwerp, Belgium

*Correspondence to*: Mahdi Nakhavali (m.nakhavali@exeter.ac.uk)

**Abstract.** Current global models of the carbon (C) cycle consider only vertical gas exchanges between terrestrial or oceanic reservoirs and the atmosphere, thus not considering lateral transport of carbon from the continents to the oceans. Therefore, those models implicitly consider that all the C which is not respired to the atmosphere is stored on land, hence overestimating the land C sink capability. A model that represents the whole continuum from atmosphere to land and into the ocean would provide better understanding of the Earth's C cycle and hence more reliable historical or future projections. We present an original representation of Dissolved Organic C (DOC) processes in the Joint UK Land Environment Simulator (JULES-DOCM). The standard version of JULES represents energy, water and carbon dynamics between vegetation, soil and atmosphere, while lateral fluxes only account for water run-off. Here we integrate a representation of DOC production in terrestrial ecosystems based on incomplete decomposition of organic matter, DOC decomposition within the soil column, and DOC export to the river network via leaching. The model performance is evaluated in five specific sites for which observations of soil DOC concentration are available. Results show that the model is able to reproduce the DOC concentration and controlling processes including leaching to the riverine system which is fundamental for integrating terrestrial and aquatic ecosystems.



## 1   Introduction

An estimated 1.9 Pg C yr$^{-1}$ is exported from soils through the river network to the oceans, which represents a significant flux
in global carbon (C) cycle (Cole et al. 2007; Regnier et al. 2013) and can affect biological and chemical properties of both
aquatic (Aitkenhead & Mcdowell 2000) and terrestrial ecosystems (Kalbitz 2000). In land surface models that are part of Earth
system models, only vertical fluxes of carbon between land and atmosphere are considered whilst lateral export fluxes are not
included. This leads to an overestimation of soil organic C (SOC) sequestration and terrestrial C sinks (Janssens et al. 2003;
Jackson et al. 2002). Hence we need to move towards a boundless C cycle model which accounts for lateral fluxes and thus
produces more accurate projections of atmospheric $CO_2$ concentrations and C stocks (Battin et al. 2009).  One of the lateral
fluxes that has been neglected is the transfer of carbon from terrestrial to aquatic ecosystems in the form of dissolved organic
C (DOC), which has been shown to be increased by anthropogenic perturbation such as land use change such as deforestation
and increased atmospheric $CO_2$ concentrations (Regnier et al. 2013). DOC contributes about 37% of the global riverine carbon
exports to the coast (Meybeck 1993) and adds to the net-heterotrophy of inland waters and related $CO_2$ emission fluxes to the
atmosphere.

The main sources of DOC in terrestrial ecosystems are plant residues (Khomutova et al. 2000), humus and root exudates
(Kalbitz et al. 2000; Van den berg et al. 2012; Marschner 1995). DOC within the soil can be the product of in-situ production
or be brought in by advective fluxes with soil water transport. It has been hypothesized that loss of the carbon from the soil by
leaching has to be taken into account to reasonably re-assess the terrestrial C budget of Europe (Siemens 2003). The fate of
this DOC within inland water networks, i.e. the proportion transported to the coast or respired and emitted to the atmosphere,
is the key to understanding the link to the other compartments of the Earth system (Cole et al. 2007; Battin et al. 2009).
Nevertheless, it is a difficult task to link riverine and terrestrial fluxes by empirical methods, because 1) riverine fluxes are
integrating fluxes from different land use systems (Kindler et al. 2011; Boyer & Groffman 1996) with different leaching rates
and DOC quality, 2) in-stream transformation makes it difficult to trace back terrestrial DOC sources, and 3) the difficulty to
separate natural and anthropogenic perturbation fluxes (Schelker et al. 2013; Regnier et al. 2013).

A physical-based modelling approach explicitly representing different terrestrial sources and processes involved in DOC
cycling within the soil column and DOC leaching from the soil can help overcome these difficulties. Representation of DOC
cycling within the soil column is also a major step toward simulating deep soil SOC formation (Rumpel & Kögel-Knabner
2011). Physical-based models help to understand the processes involved in soil DOC cycling and leaching as well as
biogeochemistry of SOC in general. So far several models have been developed that simulate DOC with different temporal
and spatial resolution, from 15 minutes as in SOLVEG-II (Ota et al. 2013) to monthly as in ECOSSE (Smith et al. 2010) or
RivCM (Langerwisch et al. 2015) and from site scale as in DyDOC (Michalzik et al. 2003) to global scale as in TEM
(Kicklighter et al. 2013). Some of these models represent DOC leaching, whereas others do not. Each model has its own
particular definition for carbon pools (including DOC) and DOC production processes which can be based on turnover time,



as in TERRAFLUX (Neff & Asner 2001), or based on chemical composition as in the DyDOC model (Michalzik et al. 2003). Although all these models have been evaluated, with the exception of the TEM model which was tested for arctic rivers, none of them has demonstrated its ability of representing the DOC production, processing and transport at the global scale.

In general, most of the models containing decomposition are based on first-order kinetics (Olson 1963). Frequently, models
tend to represent the top soil layer as the major source for DOC production and export (Koven et al. 2013), other studies (Rumpel & Kögel-Knabner 2011; Braakhekke et al. 2013) highlight the importance of DOC for SOC production in deeper soil layers.

Here, we present an original representation of DOC processes in the Joint UK Land Environment Simulator (JULES), and the new model branch is named JULES-DOCM. The standard version of JULES represents energy, water and carbon cycles and
exchanges with the atmosphere, but only account for water runoff, not including export of carbon from terrestrial ecosystems to the aquatic environments. JULES has been used to evaluate the global C cycle (e.g. LeQuéré, et al. 2015, Sitch et al. 2015) and its role in the Earth system, but to date lacks the critical processes of DOC production and export. The aim of this study is to include a representation of DOC produced in terrestrial soils down to 3 meters in JULES, assuming an incomplete decomposition of organic matter and its subsequent fate as DOC including i) DOC decomposition and release as $CO_2$ to the
atmosphere, and ii) DOC export to the riverine system via leaching; to test the new model in different ecosystems and to evaluate it against specific sites where soil DOC measurements were available.

## 2 Material and Methods

### 2.1 JULES model

JULES is a process-based model which represents energy, water and C cycling between vegetation, soil and atmosphere as described in Best et al. (2011) and Clark et al. (2011). Vegetation processes in JULES are represented in a dynamic vegetation model (TRIFFID), distinguishing 9 plant function types (PFTs) at the global scale: tropical and temperate broadleaf evergreen
trees, broadleaf deciduous trees, needle-leaf evergreen trees and deciduous trees, C3 and C4 grasses, and evergreen and deciduous shrubs (Harper et al. 2016).

The representation of SOC in JULES, follows the formulation of the RothC soil carbon scheme (Jenkinson et al. 1990; Jenkinson & Coleman 2008), distinguishing four carbon pools: decomposable plant material (DPM), resistant plant material (RPM), heterotrophic microbial biomass (BIO) and long-lived humified material (HUM). DPM and RPM pools receive litter
inputs directly from the vegetation due to defoliation, mortality and disturbance, the allocation to DPM or RPM depending on the PFT characteristics with higher fraction of decomposable litter provided from grasses and higher fraction of resistant litter provided from trees (Clark et al. 2011). HUM and BIO each receive inputs from the other two soil carbon pools, as a fraction of the decomposition that is not respired to the atmosphere.





### 2.2 JULES-DOCM model new features

JULES-DOCM is an extension of JULES based on version 4.4 (vn4.4 documentation in http://jules-lsm.github.io/vn4.4) , which explicitly represents DOC cycling in soils and considers DOC leaching from the soil profile. The following section
deals with the representation of DOC fluxes and processes in more details.

### 2.2.1 Soil carbon profile

SOC is specified as the main source of DOC in JULES-DOCM. In JULES v4.4, each of the four SOC pools is treated as a
single box down to 3 m, without any representation of its vertical distribution. This absence of vertical distribution has consequences in terms of simulating DOC fluxes, but also potential impacts on soil $CO_2$ fluxes, considering vertical variations of soil temperature and moisture. In JULES-DOCM, we introduce a vertical distribution of SOC for each soil carbon pool using a weighting factor $\beta_0$ :

$$\beta_{0_i} = e^{-\frac{z_i}{z_0}} \times dz_i \qquad\qquad (eq.1)$$

Where $z_0$ is the e-folding depth of C content within 1 meter of soil (i.e. depth at which SOC decreases by a factor of $e$ relative to the surface), $z_i$ is the soil depth of layer $i$, and $dz_i$ is the thickness of the soil layer.

$z_0$ is estimated from the vertical distribution of SOC within a 3 m soil profile based on the observed soil carbon profiles across
several biomes (Jobbágy & Jackson 2000):

$$\int_0^1 e^{-\frac{z}{z_0}} d_z = x \int_0^3 e^{-\frac{z}{z_0}} d_z \qquad\qquad (eq.2)$$

where $x$ is the SOC content percentage within 1 meter of soil relative to a 3 meter profile for different biomes (Jobbágy &
Jackson 2000). Values of $z_0$ for each PFT are given in Table S1.

In order to calculate the fraction of SOC in each layer, the weighting factors are normalised to sum to 1:

$$\beta_{z_i} = \frac{\beta_{0_i}}{\sum_{z=1}^{z=4} \beta_{0_i}} \qquad\qquad (eq.3)$$

For calculating the DOC production in each soil layer, the C content based on this calculation will be used (comparison of SOC percentage in 3 meters of soil for different biomes modelled versus measured values in Fig. S1).



### 2.2.2 DOC fluxes and processes

In JULES-DOCM, two new DOC carbon pools have been added, a labile and a recalcitrant DOC pool based on their decomposition rate (Aguilar & Thibodeaux 2005; Thibodeaux & Aguilar 2005). The labile pool is readily available for
decomposition in soil solution at all times and the recalcitrant pool is subject to slower decomposition rate (Smith et al. 2010). DOC produced from plant material pools (DPM and RPM) and microbial biomass (BIO) is directed to the labile pool, while DOC from humus (HUM) is directed to the recalcitrant pool. Each of these pools has a free and adsorbed (or so called locked) form, with only the free pool being subjected to decomposition and leaching.

DOC production ($F_P$) follows first-order kinetics (Olson 1963) and the flux of carbon from SOC to DOC pools ($k$ for labile or
recalcitrant) in each soil layer (i) in Kg C m$^{-2}$ day$^{-1}$($F_P$; arrows 1-4 Fig. 1) is calculated as:

$$F_{P_{k,i}} = \beta_{z_i} \times S_C \times \left(1 - e^{\left(-K_P \times F_S(S)_i \times F_T(T_{soil})_i \times F_v(v) \times D_f\right)}\right) \times e^{-\tau_{z_i}} \qquad (eq.4)$$

where $S_C$ is amount of carbon in the soil organic pool (DPM/RPM/BIO for DOC labile pool and HUM for recalcitrant pool)
in kg C m$^{-2}$, $K_P$ is DOC production rate in day$^{-1}$, $F_S(s)$ and $F_T(T_{soil})$ are respectively the rate modifiers due to moisture and temperature, which are controlling decomposition in each soil layer (i), $F_v(v)$ is the fraction of the vegetation. All units are given in Table 2. The moisture and temperature rate modifiers are based on the RothC formulations. $\tau_z$ is the empirical factor for decrease of C decomposition rates with soil depth, as recently introduced in JULES (Burke et al. 2016).

The DOC production rate is further modified by $D_f$, which considers the decrease of SOC decomposition rate as increase of
silt plus clay content (Parton et al. 1987):

$$D_f = 1 - (0.75 \times (clay + silt)) \qquad (eq.5)$$

After decomposition, carbon pools ($S_C$) are updated by the changes in each time step (daily) as follow:

$$\frac{\Delta S_{CARB,DPM}}{\Delta t} = f_{DPM}\Lambda_c - R_{DPM} - R_{DPM} - F_{DOC,DPM} \qquad (eq.6)$$

$$\frac{\Delta S_{C_{RPM}}}{\Delta t} = (1 - f_{DPM})\Lambda_c - R_{RPM} - F_{P_{RPM}} \qquad (eq.7)$$

$$\frac{\Delta S_{C_{BIO}}}{\Delta t} = 0.46\beta_R R_S - R_{BIO} - F_{P_{BIO}} + F_{BIO_{IN}} \qquad (eq.8)$$

$$\frac{\Delta S_{C_{HUM}}}{\Delta t} = 0.54\beta_R R_S - R_{HUM} - F_{P_{HUM}} \qquad (eq.9)$$





where $f_{DPM}$ is fraction of litter that is decomposable (depending on vegetation type), $\Lambda_c$ is litter fall, R is respiration from each C pool, $R_s$ is total respiration ($R_s = R_{DPM} + R_{RPM} + R_{BIO} + R_{HUM}$), $\beta_R$ depends on soil texture to account for the protective effect of small particle sizes These parameters were already present in JULES (Clark et al. 2011). In JULES-DOCM the update of

carbon pools after DOC production was added (last term of each equation, $F_P...$, defined in equation 4 above) as well as $F_{BIO_{IN}}$ the input flux from DOC to BIO pool, described below.

We assume that the decomposition of DOC pools ($F_D$) (Kg C m$^{-2}$ day$^{-1}$) also follows first-order kinetics depending on temperature and labile and recalcitrant DOC pool size as follow (arrows 5-6 Fig. 1):

$$F_{D_{k,i}} = S_{DOC_{k,i}} \times (1 - e^{(-K_{DOC_k} \times F_T(T_{soil})_i)}) \qquad (eq.10)$$

where $S_{DOC}$ is the DOC pool size ($k$ for labile or recalcitrant) in kg C m$^{-2}$ and $K_{DOC}$ is the basal decomposition rate of the free DOC ($k$ for labile or recalcitrant pool) (in day$^{-1}$) and $F_T(T_{soil})$ is the soil temperature rate modifier within each soil layer (i). Part of decomposed DOC is respired ($R_{DOC}$ in kg C m$^{-2}$ day$^{-1}$, arrow 7 Fig. 1) and the rest returns to the BIO carbon pool ($F_{BIO}$

$_{IN}$ in kg C m$^{-2}$ day$^{-1}$, arrow 8 Fig. 1) from each soil layer (i) and DOC pools (k). This proportion is controlled by a CUE parameter (Kalbitz et al. 2003) which is set to 0.5 as a default as in Manzoni et al. (2012).

Hence distribution of decomposed DOC to the BIO pool and respiration will be:

$$F_{BIO_{IN_i}} = (1 - CUE) \times F_{D_{k,i}} \qquad (eq.11)$$

$$R_{DOC_{k,i}} = CUE \times F_{D_{k,i}} \qquad (eq.12)$$

For adsorption/desorption, a constant sorption equilibrium distribution coefficient ($K_D$) is used to partition DOC in dissolved and adsorbed phases. The assumption is that DOC in the labile or recalcitrant pool is proportionally distributed between DOC

lock ($S_{DOCL}$, adsorbed DOC on the soil surface and free pools ($S_{DOC}$ in soluble phase) depending on $K_D$ at every time step from each soil layer(i) and DOC pool (k).

Hence these terms for DOC labile and recalcitrant pools in JULES-DOCM are as follow (arrow: 9 and 10, Fig. 1):

$$F_{AD_i} = S_{DOC_{k,i}} \times K_D \times \frac{BK}{\theta v_i} \qquad (eq.13)$$

$$S_{DOC_{k,i}} = S_{DOC_{k,i}} + \left(-F_{AD_i} + S_{DOC_{L_{k,i}}}\right) \qquad (eq.14)$$




$$S_{DOC_{L_{k,i}}} = S_{DOC_{L_{k,i}}} + \left( F_{AD_i} - S_{DOC_{L_{k,i}}} \right) \qquad \text{(eq.15)}$$

where $S_{DOC}$ is free labile and recalcitrant DOC pools in kg C m$^{-2}$, $K_D$ is the distribution factor (m$^3$ water kg$^{-1}$ soil), BK is bulk density (kg soil m$^{-3}$) and $\theta_v$ is the volumetric soil moisture (m$^3$ m$^{-3}$) and it is considered to be same for DOC labile and

recalcitrant pools.

DOC diffusion ($F_{Diff}$) in kg C m$^{-2}$ day$^{-1}$ between the layers is based on Fick's second law and it is the function of the diffusion coefficient (D) in m$^2$ day$^{-1}$, concentration of labile or recalcitrant DOC at different soil depths (C$_{DOC}$) in kg C m$^{-2}$ and the distance (z) between every two soil depths in m (arrow12, Fig. 1):

$$F_{Diff_i} = D \times \frac{\partial^2 C_{DOC_k}}{\partial z^2} \qquad \text{(eq.16)}$$

Leaching of the DOC is considered to occur from all 4 DOC soil layers. The top DOC is defined as the first two layers representing the first 35 cm of the soil. The lower two DOC layers represent the sub-soil from 35 cm down to 3 m. Soil leaching at the top DOC layer is dependent on the surface runoff whereas subsurface leaching is dependent on the subsurface runoff.

More information on the hydrology of model is given in Gedney & Cox (2003); Clark & Gedney 2008). Both DOC layers leaching fluxes are based on the concentration of free DOC in the soil water. Hence leaching of DOC (*L*) from the free labile and recalcitrant pool within the top- and sub-soil (T and S) in kg C m$^{-2}$ day$^{-1}$ is calculated as follows (arrow 11, Fig.1):

$$L_T = S_{DOC_h} \times \frac{R_{surf}}{\theta_{s_i}} \qquad \text{(eq.17)}$$

$$L_S = S_{DOC_h} \times \frac{R_{sub}}{\theta_{s_i}} \qquad \text{(eq.18)}$$

where $S_{DOC_h}$ is the DOC quantity in the free labile and recalcitrant pool (*h* for top or sub soil), $R_{surf}$ is the surface runoff, $R_{sub}$ is the subsurface runoff (both kg m$^{-2}$ day$^{-1}$) and $\theta_s$ is the soil moisture in each soil layer (i) (kg m$^{-2}$).

Hence free and locked DOC pools are updated as follow:

$$\frac{\Delta S_{DOC_k}}{\Delta t} = F_P \pm F_{AD} + F_{Diff} - F_D - L_T - L_S \qquad \text{(eq.19)}$$

Values of the main DOC model parameters are given in Table 1.




### 2.3 Sites description

Two data levels were provided in order to test the model performance. Level 1, including Hainich, Carlow and Brasschaat which included the carbon fluxes and continuous DOC measurements from soil water from 3 to 10 years period, and level 2,

including Turkey Point 89 (TP89) and Guandaushi with fewer C fluxes measurements and discontinuous DOC measurements (Table 3). Location of sites are given in Figure 2.

### 2.3.1 Hainich

The site "Hainich", located in Germany – National park Hainich, (51°04′ 45″N, 10°27′07″E), is covered by an old-growth deciduous forest dominated by *Fagus sylvatica* and intermixed with *Fraxinus excelsior* and *Acer pseudoplatanus* (Mund et al. 2010). The soil class at this site is Eutric Cambisol with a high clay content and high biological activity, as illustrated by a mull or F-Mull organic layer (Table 4). The mean annual air temperature is 7.5-8°C and the annual precipitation is in the range of 750-800 mm yr$^{-1}$ (Kutsch et al. 2010). At this site, soil solution samples were taken at three depths (5, 10 and 20 cm) at a

bi-weekly interval, applying four tension lysimeters at different plots on the site

### 2.3.2 Carlow

The site "Carlow" is located in Ireland – County Carlow, (52° 52'N, 6° 54'W). The land cover is grassland, the soil class is
Calcic Luvisol. This sandy loamy soil has a uniform profile and is well-drained (Table 4). The climate is characterized by a mean annual air temperature of 9.3°C and a mean annual precipitation of 823 mm yr$^{-1}$ (Walmsley et al. 2011). DOC samples were collected from at two locations separated 150 m from each other, using 20 suction cups per location, with ten of these cups installed directly beneath the rooting zone and the other ten at a depth of 0.7 m (Walmsley 2009).

### 2.3.3 Brasschaat

The site "Brasschaat" is located in Belgium and covered by mixed coniferous/deciduous (De Inslag) forest, (51°18'33" N, 4°31'14" E) with stands of old Scots Pine (*Pinus sylvestris)* (Janssens et al. 1999). The temperate maritime climate is characterized by a mean annual air temperature of 11.1°C and a mean annual precipitation of 824 mm yr$^{-1}$ (Gielen et al. 2010).
The soil class was defined as Albic Hypoluvic Arenosol (Table 4). The profile usually exhibits a high soil moisture, but due to the sandy texture and rapid hydraulic conductivity in upper horizons, it is rarely saturated (Gielen et al. 2011).
DOC samples were collected at three horizons of Al/Ap, A/E and Cg (10,35 and 75cm), by means of tension lysimeters on a biweekly interval. Samples were collected at three locations and pooled into one composite sample per layer for analysis (Gielen et al. 2011).



### 2.3.4 Turkey Point 89

The site "Turkey Point 89 (TP89)", located in southern Ontario – Canada, (42°77′57″N, 80°45′09″E), is covered by an evergreen needleleaf forest dominated by Eastern White pine *(Pinus strobus* L.*)* mixed with few stands of Oak, Paper birch,

Wild black cherry and Red pine (Peichl & Arain 2006) established in 1989 on agricultural lands (Peichl, Brodeur, et al. 2010). The mean annual air temperature is 8.1°C and mean annual precipitation is 832 mm yr$^{-1}$ (Peichl & Arain 2006). The soil class at this site is Gleyed Brunisolic Luvisol and due to the high sand content, it is well drained and has a low to moderated water holding capacity (Peichl, Brodeur, et al. 2010; Presant, E.W., Acton 1984). DOC sampling was attempted in monthly intervals at three depths of 25, 50 and 100 cm by means of porous cup suction lysimeters, however, due to the dry sandy soils samples

could only be retrieved for 5 separate days of sampling in 2004 and 2005 (Peichl et al. 2007).

### 2.3.5 Guandaushi

The site "Guandaushi" is located in central Taiwan, (23° 8'N, 120° 8'E). The climate is characterized by distinct rainy and dry

seasons and a mean annual air temperature of 22.4°C and annual precipitation in the range of 2300 to 2700 mm yr$^{-1}$. The land cover is subtropical mixed hardwood forest including three stands of natural hardwood and secondary hardwood on light loam textured soil and Chinese fir *(Cunninghamia lanceolate)* on heavy clay textured soil. DOC samples were collected at three depths of 15, 30 and 60 cm in three locations at bi-weekly interval by means of porous cup ceramic tension lysimeters.

### 2.4 Model input and setting

Model performance was tested against observed data from Guanduashi and four FLUXNET sites (Hainich, Carlow, Brasschaat and Turkey Point-89). In addition, the FLUXNET data base provides meteorological data for each site that could be used as forcing for simulations in JULES, while WATCH data (Weedon et al. 2010) was used as forcing for Guandaushi site. The

meteorological forcing parameters include the downward shortwave and longwave radiation at the surface (W m-2), rainfall (Kg m-2 s-1), snowfall (Kg m-2 s-1), wind speed (m s-1), atmospheric temperature (K), atmospheric specific humidity (kg kg-1) and air pressure at the surface (Pa) (Best et al. 2011).

For Brasschaat, additional model parameters such as bulk density and clay content were taken from Janssens et al. (1999). The model was run in analytical spin-up, looping 300 times over period 1996 to 2014 until all the soil variables reached a steady

state. For Hainich, site parameters were taken from Kutsch et al. (2010). The spin-up was run looping 300 times over the years 2004 to 2014. For Carlow, site parameters were taken from Walmsley (2009) and Kindler & Siemens (2010). The spin-up was run looping 300 times over the years 2004-2009. For Turkey Point-89, site parameters were taken from Peichl & Arain (2006) and spin-up was run looping 300 times over the years 2002-2007. For Guandaushi, site vegetation parameters were taken from





Liu & Sheu (2003) and soil parameters from HWSD global data and spin-up was run looping 300 times over years 1990 to 2000.

**2.5 Sensitivity test**

In order to test the sensitivity of DOC related model parameters, simulations were performed with varying values for $\beta_z$, $\tau_z$ and DOC controlling parameters such as $K_{DOC\ (labile)}$, $K_{DOC\ (recalcitrant)}$, $D_f$, $CUE$, $K_D$ and D (Table 1).

In total, 16 runs were performed by modifying each parameter once by increasing it 50% and once by decreasing it by 50%. In order to do the comparison with measurements, runs were performed for 3 meters soil depth for the periods that

measurements were available. Hence, Brasschaat runs were performed for the years 2006-2010, Hainich runs for the years 2005-2014 and Carlow runs for the years 2006-2008.

**2.6 Statistical analysis**

In order to test the model performance, with regard to simulated C stock and fluxes, we used an ANOVA (Analysis of variance) test to compare the model results from the default set of parameters against measurements. In order to test the parameter impact on the simulated DOC concentrations, we computed the RMSE values from each set of model parameter configurations.

**3 Results**

**3.1 Carbon concentration and fluxes**

To examine the performance of soil DOC simulations, it is first necessary to explore other carbon fluxes which link to soil DOC pools. The first flux to be validated is the gross primary production (GPP), for which we have observed values (Table

3). The modelled mean GPP for Brasschaat and Carlow was significantly lower than measurements with 867±25 g C m$^{-2}$ year$^{-1}$ compared to 1173.3±91 g C m$^{-2}$ year$^{-1}$and 903.2 g C m$^{-2}$ year$^{-1}$ compared to 1165.3 g C m$^{-2}$ year$^{-1}$ ($p$ <0.05, Table S2), respectively. For Turkey Point 89 and Hainich, the measured GPP was in line with our model results with 1731.5±108 g C m$^{-2}$ year$^{-1}$ and 1606.74±101 g C m$^{-2}$ year$^{-1}$ compared to 1635.1± 62 g C m$^{-2}$ year$^{-1}$ and 1455±167 g C m$^{-2}$year$^{-1}$ ($p$ = 0.162, Table S2). The modelled NPP was higher than observed values for Hainich and for Turkey Point-89, while it was lower than observed

values for Brasschaat (Table 5).

Total soil respiration measurements were available for Brasschaat, Hainich and Turkey Point-89 (Table 3) and were compared with the modelled outputs. The simulated values were close to observed values at Hainich, while the modelled values for





Brasschaat were significantly higher (p-value < 0.05, Table S2) and for Turkey Point-89 higher (p-value = 0.0896, Table S2), than the observed values (Table 5).

Finally, we compared the SOC in measurements and model outputs, where the measurements from Brasschaat for 100 cm, Hainich for 60 cm, Carlow for 50 cm and Turkey Point-89 for 15cm (A-horizon) of soil were available. The modelled SOC

stock for Brasschaat in the first 100 cm and for Hainich down to 60 cm were slightly lower than the observations, while for Carlow the simulated stocks down to 50 cm and for Turkey Point-89 the simulated stocks down to 15 cm were higher than the observed stocks (Table 5).

### 3.2 DOC simulations

In general, JULES-DOCM was capable of reproducing the DOC concentrations at all the tested sites using the default set of parameters (Table 1) chosen as representative for the top soil (Fig. 3 Level 1 sites, Fig. 4 level 2 sites). For Hainich, the simulated average values and value range were close to observed values at 10 cm and 20 cm (Table 5, RMSE values for 10 cm and 20 cm are 3.0 and 2.5 mg $L^{-1}$ respectively). For Brasschaat, the simulation underestimated DOC concentrations at all

depths, but with an increasing underestimation with soil depth (Table 5, RMSE values for 10, 35 and 75 cm are 22.9, 18.4 and 16.8 mg $L^{-1}$ respectively). For Carlow, the modelled and measured values were close at depths of 10 cm and 77 cm, but strongly underestimated at the intermediate depth of 28 cm (Table 5, RMSE values for 10, 10-38 and 28-77 cm are 3, 10.2 and 1.5 mg $L^{-1}$ respectively). At Turkey Point-89, the modelled and observed values were close at 25cm depth, but the DOC concentration average over the profile down to 100 cm was overestimated (Table 5). For Guandaushi, DOC measurements from three

different stands (Natural hardwood, secondary hardwood and Chinese fir) values were compared with modelled values. The model values for a depth of 15 cm were closer to observed values for Chinese fir than for natural hardwood or secondary hardwood sites. For 30cm depth, the simulated DOC concentration was substantially lower than the measured DOC averaging over three stands in Guandaushi (Table 5).

Overall, the model was capable of reproducing the seasonality of DOC concentrations for the European sites where long-term

observation data are available (Fig. 5). However, at Braschaat the simulated DOC peaked from April-July while observed DOC peaked from July-September.

We also examined the hydrology of the model and its interaction with DOC concentration and leaching (e.g. Hainich - Fig. 6; other sites are plotted in Fig. S4). It can be seen for the period 2005 to 2014 that during heavy precipitation, high runoff was produced which caused the higher leaching, and the consequence was a drop in the DOC concentration in 3 meters of soil.



### 3.3 Sensitivity tests

Sensitivity to model parameters was tested on the three European sites where a representative time-series of observed DOC concentrations was available (e.g. Hainich-10cm, Fig. 7). The results indicate that among all the parameters in all three sites,

the model shows the highest sensitivity to SOC vertical profile, controlled by parameter $\beta_z$ (eq. 3), and the changing of SOC decomposition rate with soil depth, parameter, $\tau_z$ (eq. 4) (p-values < 0.05, Table S5). Among the DOC controlling parameters, the model shows the highest sensitivity to the basal decomposition rate of recalcitrant DOC ($K_{DOC \ (recalcitrant)}$) (eq.10), which is the inverse of the residence time of DOC in the recalcitrant pool.

The sensitivity of the model to each of these parameters was different at each site. For Hainich, the highest sensitivity was

assigned to $\tau_z$. Here, a change in $\tau_z$ by 50% leads to a 36% change in the mean DOC within 3 m, while a 50% change in $K_{DOC \ (recalcitrant)}$ leads to a 29% change and a 50% change in $\beta_z$ leads to a 25% change in simulated DOC concentrations (Fig 8a). The closest value for the mean DOC in 10 cm in Hainich (8.8 mg L$^{-1}$) to the measurement was produced by the default set (8.9 mg L$^{-1}$), while the highest value for DOC was reached with the 50% increase in $\tau_z$ (12.7 mg L$^{-1}$) and the lowest DOC value was produced with 50% decrease in $\tau_z$ (4.7 mg L$^{-1}$). In contrast to that, at a depth of 20 cm, the closest value to the mean of

measured DOC (5.6 mg L$^{-1}$) was produced by 50% decrease in $K_{DOC \ (recalcitrant)}$ (4.9 mg L$^{-1}$) (Fig. 9-a).

In Brasschaat, the highest sensitivity was to $\beta_z$, closely followed by $\tau_z$ and $K_{DOC \ (recalcitrant)}$. A 50% change in each of these parameters led to a 36-40% change in DOC concentration over the 3 meters of soil profile (Fig. 8-b). At 10 cm, the closest value to measurements mean (39.4 mg L$^{-1}$) was produced by 50% increase in $\tau_z$ (39.2 mg L$^{-1}$). At 35 cm depth, the closest value to mean measurement (29.3 mg L$^{-1}$) was calculated by 50% increase in $K_{DOC \ (recalcitrant)}$ (16.2 mg L$^{-1}$) which was also the

highest simulated value as well. At 75 cm, the closest value to mean of DOC measurement (22.0 mg L$^{-1}$) was produced by 50% increase in $K_{DOC \ (recalcitrant)}$ (8.1 mg L$^{-1}$) as it was the highest of the simulated values (Fig. 9-b).

For Carlow, the most sensitive parameters were $\tau_z$ and $K_{DOC \ (recalcitrant)}$: a 50% change in those parameters leads to a 31.5% and 27.4% in simulate DOC. A 50% change in $\beta_z$ leads to a low but still significant change of 6.5% change in simulated DOC within 3 meters of soil (p-value <0.05, Table S5) (Fig. 8-c). In 10cm, the closest modelled value to the mean measurement (5.7

mg L$^{-1}$) was produced by default parameter set (5.8 mg L$^{-1}$). Between 10 to 28 cm all the parameter sets underrepresented the DOC concentration mean measurement (13.1 mg L$^{-1}$) and the closest and highest value was produced by 50% in $\tau_z$ (3.8 mg L$^{-1}$). For 28 to 77cm, the closest value to the measurement (4.8 mg L$^{-1}$) was calculated by increasing $\tau_z$ by 50% (4.5 mg L$^{-1}$) (Fig. 9-c).



## 4 Discussion

### 4.1 Carbon concentrations, stocks and atmospheric exchange

Overall, JULES-DOCM reproduced the range of GPP for most of our sites to an acceptable degree. At some sites, due to over/underestimated autotrophic respiration, the NPP and total respiration values were slightly different than measurements. Consequently, the modelled carbon stocks were different from the measurements in most of the sites, but yet capable of representing the general patterns that were observed in the measurements.

In Brasschaat, the modelled SOC was lower than the measurements, which could be due to the underestimated NPP (Table 5)

and, as a consequence, the underestimated litter input, but also due to the overestimated soil respiration and SOC decomposition rates. In Hainich, a slightly overestimated NPP partly counter-balanced the overestimated soil respiration. Nevertheless, the SOC concentration simulated down to 60 cm was lower than the measurement at this depth. As we did not have observations of SOC down to 3 meters, we cannot certainly say if the simulated total SOC stock (13.7 Kg C m$^{-2}$) over the whole soil column is close to the reality or not. In Carlow, the slight overestimation of GPP led to the overestimated SOC concentrations down

to 50 cm, whilst again we cannot say with certainty that the whole SOC stock is overestimated, as the SOC stock has not been measured down to three meters. Some sources suggest that the SOC in Carlow grassland could be higher than the reported value in our reference, if we calculate the C in soil based on the fraction of loss of ignition (LOI) (Walmsley 2009; Hoogsteen et al. 2015). As Carlow is our only grassland biome site, additional data from different study sites would be valuable to achieve a more representative parametrization of soil carbon processes under grass land. One of the parameters to be optimized for

such sites could be CUE which has a strong impact on the stocks and fluxes. Also, since the measured values for NPP or soil respiration for this site were not available to us, we were unable to assess whether we over- or underestimated these fluxes and if this could have potentially biased our SOC stock simulations.

At Turkey Point-89, the simulated GPP is close to the observations, while NPP is slightly overestimated. The simulated soil respiration and decomposition rates are higher than observed values. The overestimated SOC concentration in the top soil

could be the result of an overestimated depth gradient in SOC concentration, which in our simulations is derived from global data (Jobbágy & Jackson 2000). Also, we simulated the steady state SOC profile for forest vegetation, whereas the forest stand at the site is relatively young and succeeded agricultural land use in 1989, and thus, the SOC profile is likely not representative for a forest site. In Gundaushi due to the lack of SOC, or vegetation carbon fluxes measurements from the site, we have no information on SOC concentrations and stocks.



### 4.2 DOC simulation

Some of the controlling parameters like DOC basal decomposition rates are kept constant over the soil profile in our simulation, while they are maybe not constant with depth in the real world, perhaps due to priming effects (Guenet et al. 2010). That could

explain why at Hainich, the simulated and observed DOC concentrations are very close at 10 cm depth, while they differ more at 20 cm depth. In Brasschaat, the underestimation of SOC as a source of DOC led to a general underestimation of DOC. Nevertheless, the decrease of relative DOC concentration through soil is consistent with the observations.

In Carlow, the measurements were provided from two plots which were placed on different terrain positions. The measurements from plot 2 (150 meter in south-westerly direction from plot 1at 10 to 28 cm depth had a higher DOC

concentration than plot 1 at the 10 cm (Walmsley 2009). This could be the result of small scale variations related to terrain position, which can be related to different soil moisture regimes and lateral import of DOC. It is not possible to represent such small-scale variation in global models like JULES-DOCM.

At Turkey Point-89, the overestimated DOC concentration for 100 cm depth may be due to a change in land use, which was not taken into account during simulations, consequently providing more C input for DOC production as mentioned above. At

this site, the observed higher soil moisture in the deeper profile could indicate a potentially high advection of DOC to the lower layers (Peichl, Arain, et al. 2010).  This could be another reason for the lower DOC in 100 cm from measured compared to the modelled results.

In Gundaushi, the lower values of DOC from our model compared to the measurements could be due to: Firstly, the high temporal variability of observed concentrations (large standard deviation for all the depths from the three stands). Second, the

high value of DOC input from rainfall, which is not represented in JULES-DOCM (Liu & Sheu 2003). Recent studies have indicated that including this flux in models can have a significant impact on the DOC in soil (Lauerwald et al. 2017).

As there are no measurements of lateral leaching of DOC from soil to the river, our evaluation of this flux is based on the simulated DOC concentration and runoff. Hence as the simulated hydrology of the JULES model has been evaluated previously (Gedney, N. , Cox 2003; Clark & Gedney 2008), in this study, we assume that we will get robust estimates of DOC leaching

by multiplying simulated concentration by runoff, as long as simulated DOC concentrations can be validated.

Overall, besides over/underestimation of DOC at some sites, the model was capable of representing the trend of DOC concentration at different depths when comparing to the measurements at all the sites.



### 4.3 Sensitivity analysis

The sensitivity tests indicate that the parameters controlling SOC concentrations in the soil profile and the recalcitrant DOC residence time have the most significant effect on soil DOC concentration, which indicates the importance of factors

controlling DOC sources Nevertheless, DOC related model parameters such as basal DOC decomposition rate are constant over different depths, which could be the reason for the difference between the modelled and measured values, especially in the deeper soil layers. Hence, it is important to introduce a depth-dependence decay rate for these parameters.

A limitation in our simulation is that we use a single, calibrated value for recalcitrant DOC residence time, which is the most sensitive DOC controlling parameter. It has been shown that this parameter can vary with biodegradability of SOC and litter

under different PFTs and at different sites (Kalbitz et al. 2003; Turgeon 2008). However, more detailed data for different biomes is needed for calibrating different residence times for different PFTs.

### 5 Conclusion

Applying a carbon cycle model that integrates the whole continuum from land to ocean to atmosphere provides a better understanding of the Earth's carbon cycle and makes more reliable future projections. In this study, we presented DOC related processes in JULES, JULES-DOCM, which includes the DOC produced in the soil down to three meters and its subsequent fate including its decomposition and release as $CO_2$ to the atmosphere, and its export to the river network via leaching in different ecosystems. Results show that the model is capable of representing the DOC stocks, processes and its export to the

riverine systems from different ecosystems. In future, our developments in the representation of DOC leaching will lead to a model approach integrating terrestrial and aquatic C cycling. However, more field data are still required to improve the model parametrization and performance.

### Code availability

The code written for this version of JULES can be found at:

https://code.metoffice.gov.uk/svn/jules/main/branches/dev/mahdinakhavali/vn4.4_JULES_DOCM/ (registration required)

*Acknowledgments:* The research leading to these results has received funding from the European Union's Horizon 2020 research and innovation programme under the Marie Sklodowska-Curie grant agreement No 643052 (C-CASCADES project). We want to thank Altaf

Arain, Tim Moore and Gerd Glexiner for providing the DOC measurements. Ronny Lauerwald received funding from the European Union's Horizon 2020 research and innovation program under grant agreement no. 703813 for the Marie Sklodowska-Curie European Individual Fellowship "C-Leak". Jing Tang is financed by Marie Sklodowska-Curie Action Individual Fellowship (MABVOC: 707187) and supported by Danish National Research Foundation (CENPERM DNRF100). Marta Camino-Serrano acknowledges funding from the European Research Council Synergy grant ERC-2013-SyG-610028 IMBALANCE-P.





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



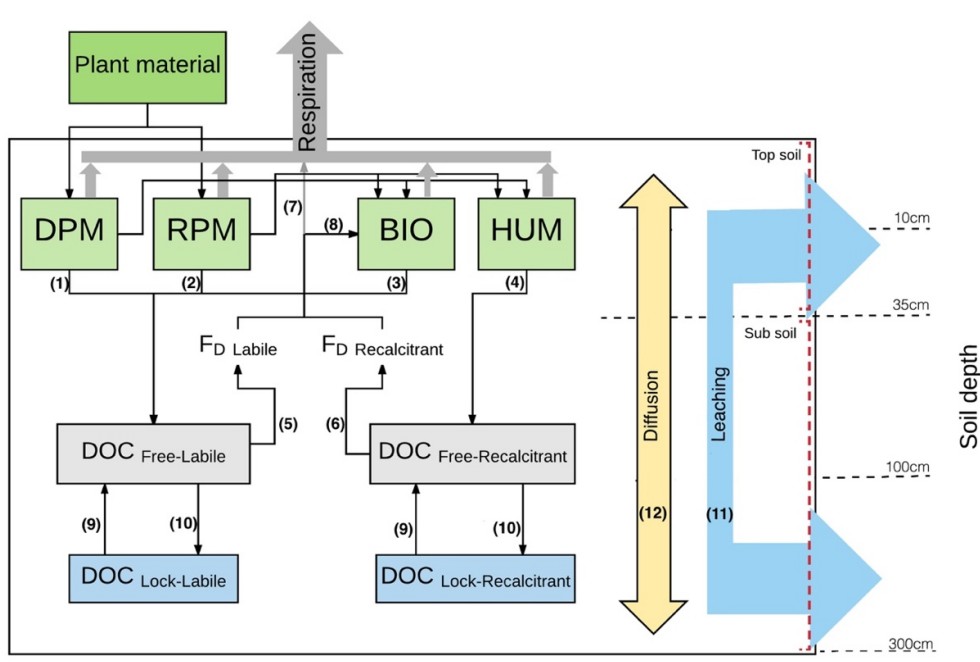

Figure 1. JULES-DOCM model structure





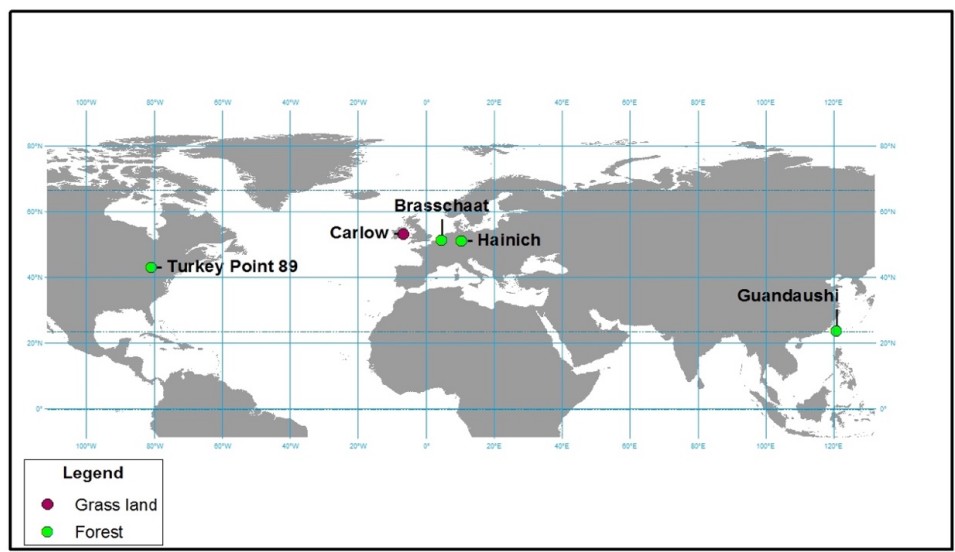

Figure 2. Study sites





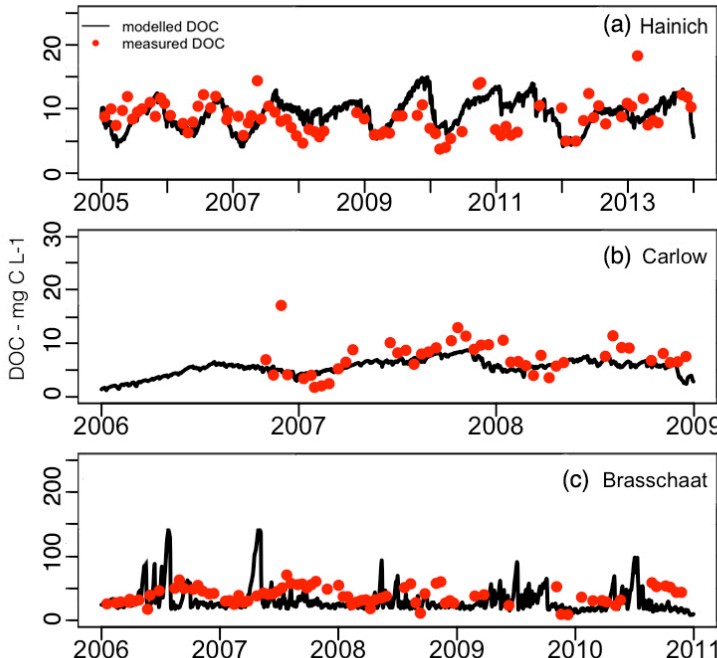

5   Figure 3. DOC concentration (mg C L$^{-1}$) at 10 cm depth measured (red dots) and simulated (black lines) for (a)
Hainich, (b) Carlow, and (c) Brasschaat. Results for other depths are given in Figure S2.





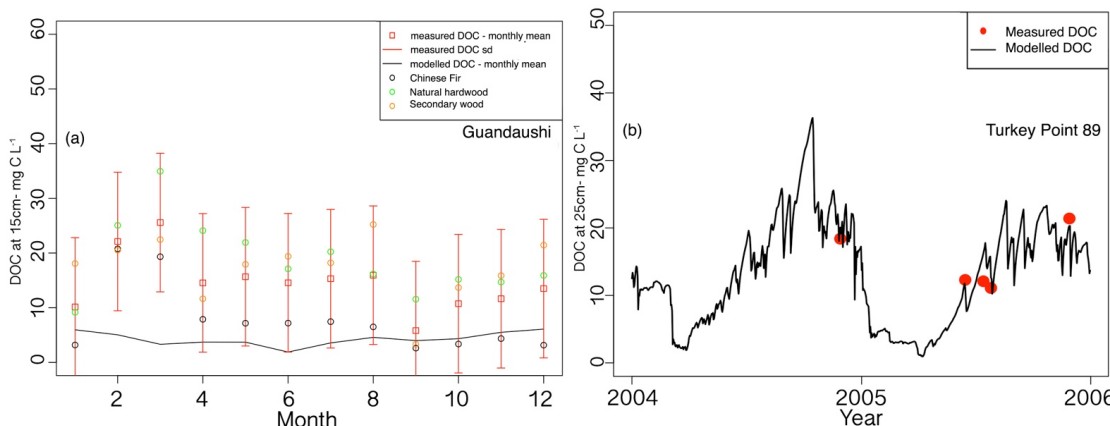

Figure 4. DOC concentration (mg C L$^{-1}$) for (a) Guandaushi at 15 cm measured (black dot: Chinese Fir, green dot: natural hardwood, orange dot: secondary wood, red square: mean, red line: standard deviation) and simulated (black lines) and for (b) Turkey Point 89 at 25 cm measured (red dots) and simulated (black lines). Results for other depths are given in Figure S3.





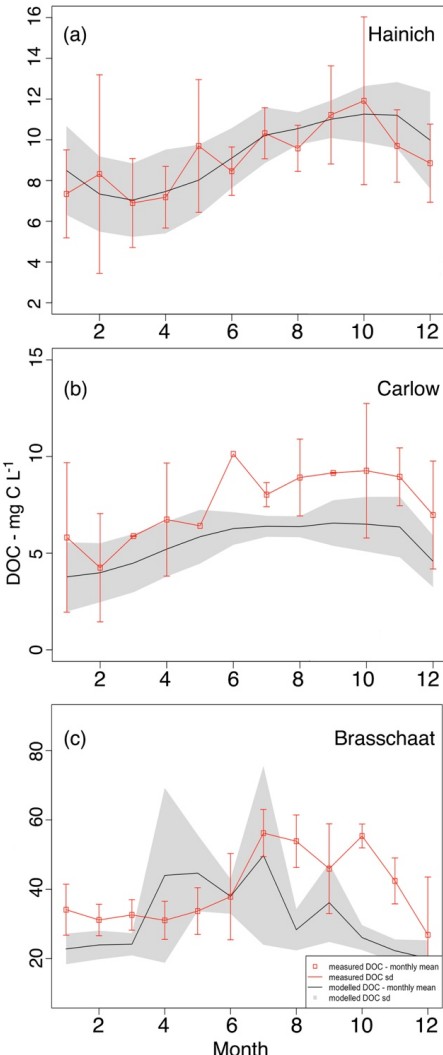

Figure 5. a) Monthly DOC (mg C L$^{-1}$) at 10 cm in Level 1- sites modelled (black line: mean, grey line: standard deviation) versus measured (red square: mean, red line: standard deviation) for studied period (a) Hainich averaging from 2005-2014 (b) Carlow, averaging from 2006-2008 (c) Brasschaat, averaging from 2006 –2010. Results for other depths are given in Figure S7.



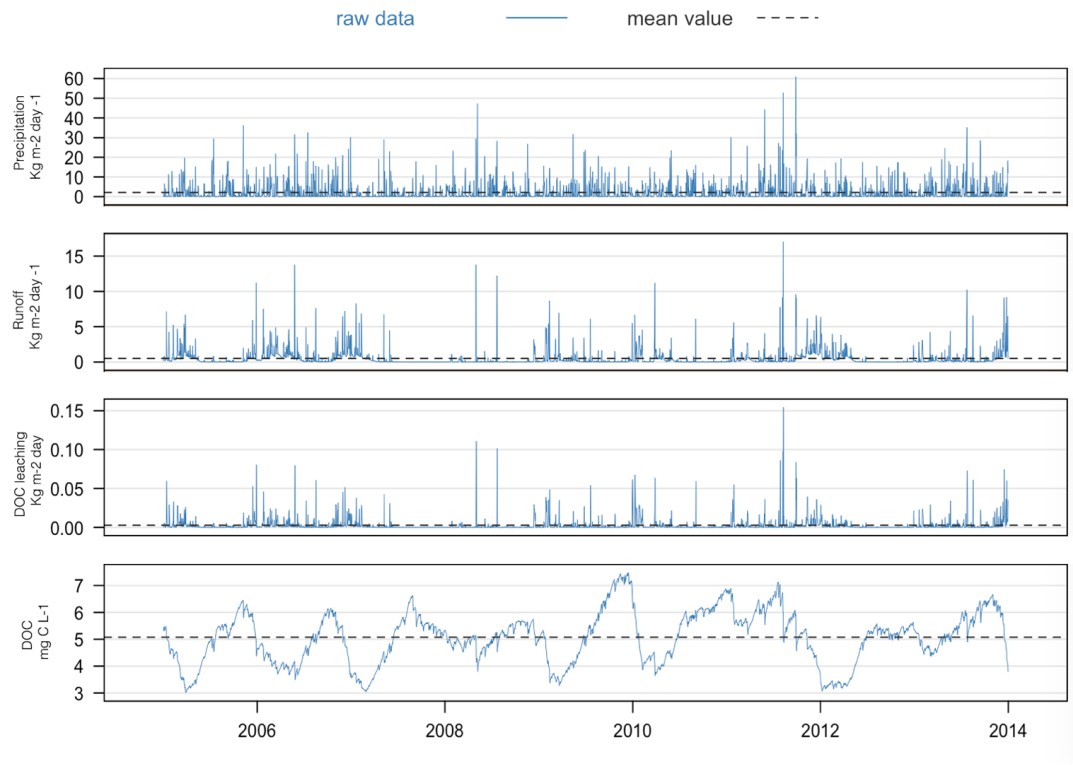

Figure 6. Observed precipitation, simulated runoff, DOC leaching and DOC concentration in Hainich from 2006 to 2013 indicating the relation between the averaged DOC concentrations at 3 m of soil with leaching as a result of runoff that follows large precipitation events.





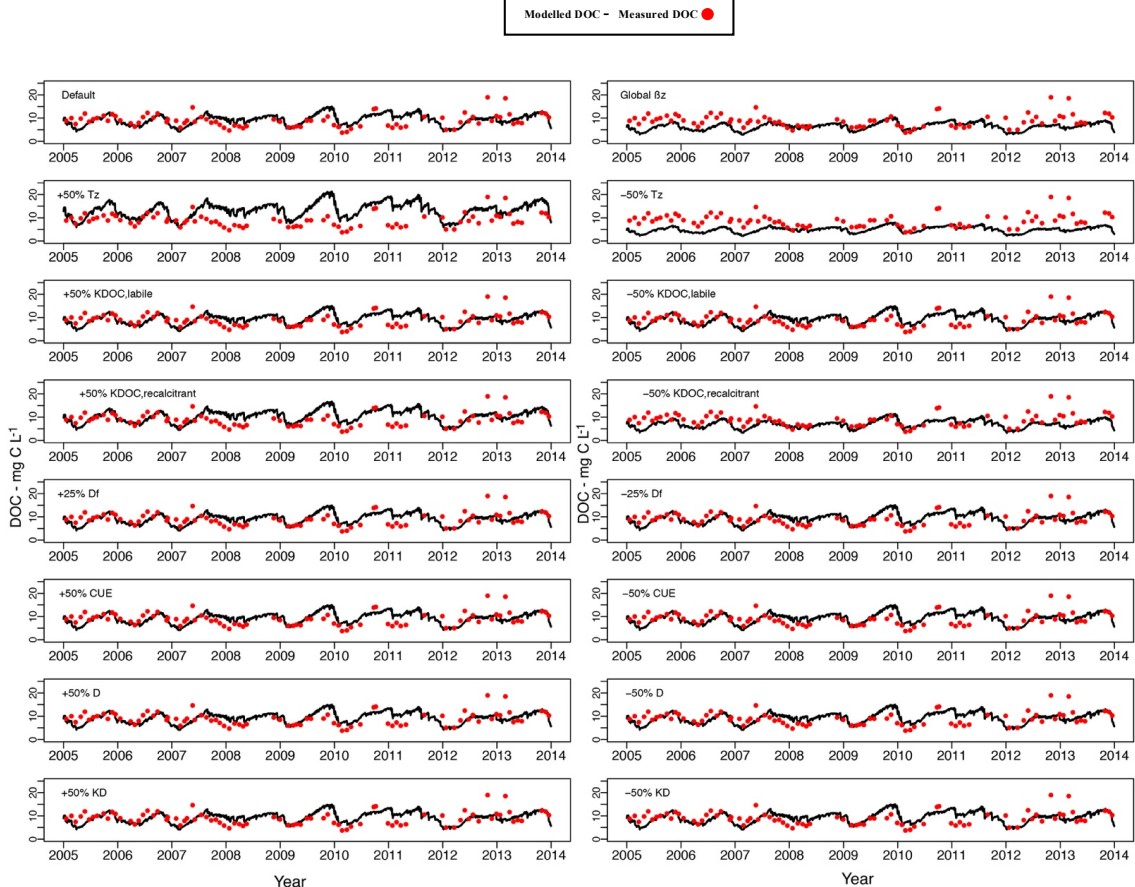

Figure 7. DOC concentration (mg C L$^{-1}$) simulated with sensitivity parameter sets (black line) versus measured (red dot) at 10 cm depth in Hainich for period 2004-2013. Parameter sets description and values are given in Table 1. Results for other sites are given in Figure S2.





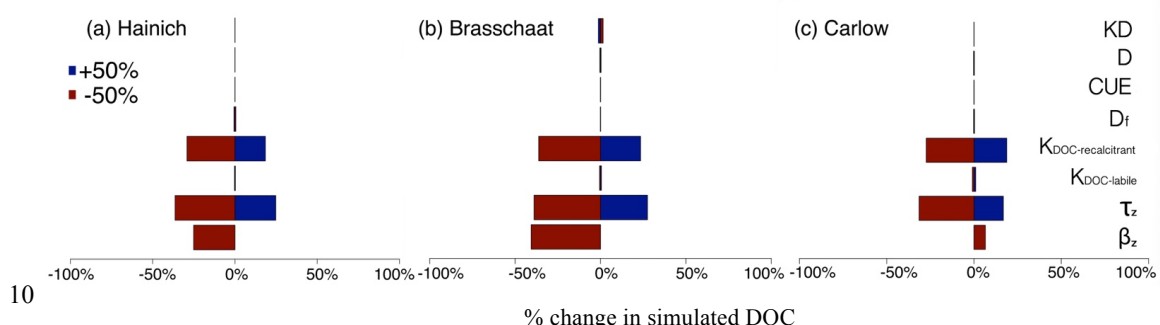

Figure 8. Relative change in simulated DOC (%) for a +50% (blue) and -50% (red) change in each parameter for level 1- sites: (a) Hainich, (b) Brasschaat and (c) Carlow. Values are given in Table S4.



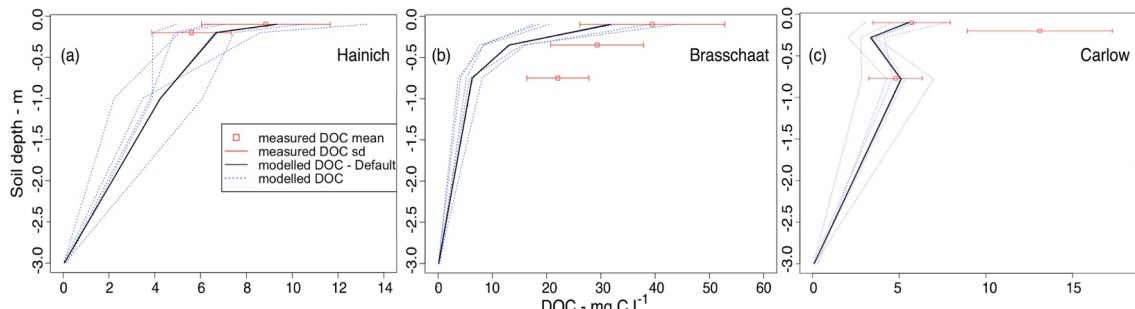

Figure 9. DOC concentration (mg C L$^{-1}$) in 3 m soil depth at level 1-sites modelled (black line: default parameter set; blue dashed line: sensitivity test parameter set) vs. measured (red square: mean; red line: standard deviation)

10    for (a) Hainich (b) Brasschaat (c) Carlow. Plot of each parameter in 3 m soil depth in Figure S6.





Table 1. DOC relevant parameters in JULES-DOCM model

| Parameter | Description | Value | Unit | Sensitivity test values (+/-) | |
|---|---|---|---|---|---|
| **Carbon parameters** | | | | | |
| $\beta_z$ | Carbon distribution with depth, depending on biome[1] | Values range (65.68 - 167.13) | $m^{-1}$ | PFT based | 109.55 |
| $\tau_z$ | Decay of Carbon decomposition with depth (z)[2] | 2 | $m^{-1}$ | 3.0 | 1.0 |
| **DOC parameters** | | | | | |
| $K_P$ | Rate constant for DOC production specific to each carbon pool[3] | 1e-4, 5e-6, 5e-5, 2e-6 | $day^{-1}$ | - | - |
| $K_{DOC\ (labile)}$ | Basal decomposition rate of free DOC labile pool[4] | 3.0 Value range (0.46-100) | days | 4.5 | 1.5 |
| $K_{DOC\ (recalcitrant)}$ | Basal decomposition rate of free DOC recalcitrant pool[5] | 600.0 Value range (66-5000) | days | 900.0 | 300.0 |
| $D_f$ | DOC production/decomposition modifier depending on clay and silt fraction[6] | 0.75 | - | 1.0 | 0.5 |
| CUE | Carbon use efficiency[7] | 0.5 | - | 0.75 | 0.25 |
| $K_D$ | Distribution coefficient of adsorbed DOC [8] | 8.05e-6 | $m^3$ water $Kg^{-1}$ soil | 1.207e-4 | 4.025e-6 |
| D | DOC diffusion coefficient [9] | 1.062e-05 | $m^2\ day^{-1}$ | 1.594e-05 | 5.313e-06 |

[1] Jobbágy & Jackson 2000

[2] Koven et al. 2013; Burke et al. 2016

[3] Jo Smith et al. 2010

[4] Kalbitz et al. 2003; Turgeon 2008

[5] Kalbitz et al. 2003; Turgeon 2008

[6] Parton, W et al. 1987

[7] Manzoni et al. 2012

[8] Moore et al. 1992

[9] Ota et al. 2013





Table 2. Symbols definition and units

| Symbol | Units | Definition |
|---|---|---|
| $BK$ | Kg m$^{-3}$ | Bulk density |
| $\beta_R$ | | Fraction of soil respiration |
| $\beta_z$ | m$^{-1}$ | Carbon distribution with depth, depending on biome |
| CUE | | Carbon use efficiency |
| $C_{DOC}$ | Kg C m$^{-2}$ | Amount of DOC subjected to transport by diffusion |
| D | m$^2$ day$^{-1}$ | DOC diffusion coefficient |
| $D_f$ | | DOC production / decomposition modifier depending on clay and silt fraction |
| $dz$ | m | Soil layer thickness |
| $\Delta S_{C_{BIO}}$ | Kg C m$^{-2}$ day$^{-1}$ | Biomass carbon pool update |
| $\Delta S_{C_{DPM}}$ | Kg C m$^{-2}$ day$^{-1}$ | Decomposable plant material carbon pool update |
| $\Delta S_{C_{HUM}}$ | Kg C m$^{-2}$ day$^{-1}$ | Humus carbon pool update |
| $\Delta S_{C_{RPM}}$ | Kg C m$^{-2}$ day$^{-1}$ | Resistant plant material carbon pool update |
| $\Delta S_{DOC}$ | Kg C m$^{-2}$ day$^{-1}$ | Labile and recalcitrant DOC pools update |
| $F_{AD}$ | Kg C m$^{-2}$ day$^{-1}$ | Flux of adsorbed DOC from labile and recalcitrant pools |
| $F_{BIO_{IN}}$ | Kg C m$^{-2}$ day$^{-1}$ | Decomposed DOC flux from labile and recalcitrant pool into biomass pool |
| $F_D$ | Kg C m$^{-2}$ day$^{-1}$ | Labile and recalcitrant decomposed DOC flux |
| $F_{Diff}$ | Kg C m$^{-2}$ day$^{-1}$ | Flux of DOC transported by diffusion |
| $F_{P_{BIO}}$ | Kg C m$^{-2}$ day$^{-1}$ | DOC flux originated from biomass carbon pool |
| $F_{P_{DPM}}$ | Kg C m$^{-2}$ day$^{-1}$ | DOC flux originated from decomposable plant material carbon pool |





| Symbol | Units | Definition |
|---|---|---|
| $F_{P_{HUM}}$ | Kg C m$^{-2}$ day$^{-1}$ | DOC flux originated from humus carbon pool |
| $F_{P_{RPM}}$ | Kg C m$^{-2}$ day$^{-1}$ | DOC flux originated from resistant plant material carbon pool |
| $F_S(s)$ | Kg m$^{-2}$ | Soil moisture rate modifier |
| $F_T(T_{soil})$ | K | Soil temperature rate modifier |
| $F_v(v)$ | | Fractional coverage of a vegetation type |
| $f_{dpm}$ | | Fraction of litter that is decomposable plant material |
| i | m | Soil layer |
| k | | DOC pool type (labile or recalcitrant) |
| $K_P$ | day$^{-1}$ | Rate constant for DOC production specific to the pool |
| $K_{DOC}$ | days | Basal decomposition rate of free DOC labile and recalcitrant pools |
| $K_D$ | m$^3$ water Kg$^{-1}$ soil | Distribution coefficient of adsorbed DOC |
| $\Lambda_c$ | Kg C m$^{-2}$ day$^{-1}$ | Litterfall rate |
| $L_T$ | Kg m$^{-2}$ day$^{-1}$ | Leaching from labile and recalcitrant DOC pools in top soil |
| $L_S$ | Kg m$^{-2}$ day$^{-1}$ | Leaching from labile and recalcitrant DOC pools in sub soil |
| m | | DOC decomposition rate type (labile or recalcitrant) |
| $R_{BIO}$ | Kg C m$^{-2}$ day$^{-1}$ | Respiration from biomass carbon pool |
| $R_{DPM}$ | Kg C m$^{-2}$ day$^{-1}$ | Respiration from decomposable plant material carbon pool |
| $R_{DOC}$ | Kg C m$^{-2}$ day$^{-1}$ | Respiration from labile and recalcitrant DOC pools |
| $R_{HUM}$ | Kg C m$^{-2}$ day$^{-1}$ | Respiration from humus carbon pool |
| $R_{RPM}$ | Kg C m$^{-2}$ day$^{-1}$ | Respiration from resistant plant material carbon pool |
| $Rsurf$ | Kg m$^{-2}$ day$^{-1}$ | Surface Runoff |



| Symbol | Units | Definition |
|---|---|---|
| $Rsub$ | Kg m$^{-2}$ day$^{-1}$ | Sub-Surface Runoff |
| $S_C$ | Kg C m$^{-2}$ | Soil carbon storage |
| $S_{DOC}$ | Kg C m$^{-2}$ | Labile and recalcitrant DOC storages |
| $S_{DOC_L}$ | Kg C m$^{-2}$ | Adsorbed labile and recalcitrant DOC storages |
| $\theta_s$ | Kg m$^{-2}$ | Soil moisture content |
| $\theta_v$ | Kg m$^{-3}$ | Volumetric Soil moisture content |
| $\tau_z$ | m$^{-1}$ | Decay of Carbon decomposition with depth |
| $z$ | m | Soil depth |
| $z_0$ | m | e-folding depth of carbon content within 1 meter of soil |



5   Table 3. Data availability for model evaluation at different

| Sites | Brasschaat[1] | Carlow[1] | Guandaushi[2] | Hainich[1] | Turkey-point89[2] |
|---|---|---|---|---|---|
| **Carbon fluxes** | | | | | |
| *GPP* | 2000-2006 | 2008 | | 2000-2012 | 2005-2008 |
| *NPP* | 2000 | | | 2000-2007 | 2005-2008 |
| *Soil respiration* | 2000-2006 | | | 2000-2007 | 2005-2008 |
| *C content* | 1995-1998 | 2006-2009 | | 2000-2007 | 2004-2006 |
| **DOC measurements** | | | | | |
| *1 year* | | | 1999 | | |
| *1 to 5 years* | | 2006-2009 | | | 2004-2005 |
| *5 to 10 years* | 2000-2008 | | | 2001-2014 | |

1. level 1 site   2. Level 2 site





Table 4. Evaluation Level 1-sites characteristics

| | Site | | |
|---|---|---|---|
| | **Brasschaat** | **Carlow** | **Hainich** |
| | **Characteristics** | | |
| **Ecosystem** | Evergreen forest | Grassland | Deciduous forest |
| **Soil classification** | Arenosol | Luvisol | Cambisols |
| **Bulk density (Kg m-3)** | 1.4 | 1.07 | 1.2 |
| **Clay (%)** | 3.4 | 22 | 58.9 |
| **Sand (%)** | 89.12 | 51 | 3.1 |
| **Silt (%)** | 7.48 | 27 | 38 |
| | **Measurement depth (cm)** | | |
| **Carbon content** | 100[1] | 50[2] | 60[3] |
| **DOC concentration** | 10,35,75 | 5,10,20 | 10-77 |
| | **FLUXNET meteorological observations** | | |
| **Period** | 1996-2014 | 2004-2014 | 2004-2009 |

1. Janssens et al. 1999 2. Kindler & Siemens 2010 3. Schrumpf et al. 2011





Table 5. The measured (Obs.) vs. the modelled (Mod.) carbon fluxes, SOC concentration and soil DOC concentration at different soil depths in five study sites.

| Variables | Level-1 | | | | | | | | Level-2 | |
|---|---|---|---|---|---|---|---|---|---|---|
| | Brasschaat | | Carlow | | Hainich | | Turkey Point-89 | | Guandaushi | |
| | Obs. | Mod. | Obs. | Mod. | Obs. | Mod. | Obs. | Mod. | Obs. | Mod. |
| Carbon fluxes (g C m$^{-2}$ yr$^{-1}$) and SOC (Kg C m$^{-2}$) | | | | | | | | | | |
| GPP | 1173±92 | 867±25 | 903 | 1165 | 1606±102 | 1455±168 | 1732±108 | 1635±63 | - | - |
| NPP | 850 | 596.1 | - | - | 673±33 | 833±153 | 814±51 | 1013±92 | - | - |
| Soil Res* | 411±34 | 625±54 | - | - | 883±206 | 909±66 | 693±16 | 1006±142 | - | - |
| SOC | 11.47 | 8.01 | 2.3 | 4.17 | 11.75 | 8.63 | 1.85 | 3.39 | - | - |
| DOC concentration (mg C L$^{-1}$) | | | | | | | | | | |
| 10 cm | 39±15 | 28±13 | 7±3 | 6±1 | 9±3 | 9±2 | - | - | - | - |
| 15 cm | - | - | - | - | - | - | - | - | nh: 19±12 sh:17±12 cf: 8±15 | 4±1 |
| 20 cm | - | - | - | - | 6±2 | 7±2 | - | - | - | - |
| 25 cm | - | - | - | - | - | - | 15±4.5 | 16±4 | - | - |
| 10-28 cm | - | - | 13±4 | 4±1 | - | - | - | - | - | - |
| 30 cm | - | - | - | - | - | - | - | - | nh: 9±7 sh: 15±8 cf: 7±17 | 3±1 |
| 35 cm | 29±2 | 13±9 | - | - | - | - | - | - | - | - |
| 75 cm | 22±1 | 6±6 | - | - | - | - | - | - | - | - |
| 28 to 77 | - | - | 5±2 | 5±0.2 | - | - | - | - | - | - |
| 100 cm | - | - | - | - | - | - | 2.2±0.2 | 7.9±2 | - | - |

*: soil respiration, nh: Natural Hardwood; sh: Secondary hardwood; cf: Chinese fir