# Peer review of "Representation of dissolved organic carbon in the JULES land surface model (vn4.4 JULES-DOCM)"

_Geoscientific Model Development, 2017_

## Referee Comment (RC1) · Anonymous Referee #1 · 16 Sep 2017

Comment on 'JULES-DOCM: representation of dissolved organic carbon in the JULES land surface model' by Mahdi Nakhavali et al.

The authors represent a modelling development that makes it possible to have a more complete look at the global carbon cycle. They combine the well-established JULES model with a newly developed model for DOC, including soil carbon processes and leaching. This manuscript is therefore an important step towards a full carbon cycle understanding.

In general I think that the manuscript is well structured and the figures are helpful to understand the outcomes. However, there are some changes needed to make it more

convincing. While I see some issues that should be clarified/solved first, I recommend publishing the manuscript in GMD after revision.

Major comments/questions

1. Please explain why the authors did not include the production of OC in soils and rivers (i.e. aquatic photosynthesis)? 2. Explain the additional value of adding Turkey Point and Guandaushi. To me it seems that the data from these two sides do not add much information, due to their much shorter time coverage. Also in the discussion it is stated that e.g. Turkey Point is not really useful because it's located at a site that was agriculturally used until 1989. Possibly it would help to remove the Level 2-sites. 3. Discuss the different methods in the study side description. Are there fundamental differences between 'suction cups' (p.8 l.22) and 'tension lysimeters' (p.8 l.32)? Are the observed values comparable? When did this sampling take place (e.g. Turkey Point – 'samples could only be retrieved for 5 separate days'; all in summer? or winter?) (see also comment above (2.)) 4. Elaborate on the model input in more detail: a) Have the data of FLUXNET and WATCH been somewhat corrected to be comparable? b) Mention the names for the parameters as it's used in the tables and equations consistently (e.g. p.9 l.28: bulk density and clay content) 5. Please adapt the figures in a way that makes them easier to understand. a) Fig.2 The extent can be smaller to better see where the sites are. b) Fig.3 Decrease the y-axis range. A maximum of about 20 should be sufficient and the differences a better to see. c) Fig .4 Decrease the y-axis range to a maximum of 40. 6. Combine both parts of the discussion to one. This would avoid the repetitions and can clearly combine all information/discussion on each of the sites.

Minor comments:

a) Define SOC at its first occurrence (p-2 l.28) b)Please use consistent and not confusing naming for the variables/parameters in the equations (e.g. R can mean 'run-off' and 'respiration') c) Connecting the equations to the arrows in the flow chart (Fig.1)

[Figure]

would help in understanding the calculations. What do the numbers (1) to (12) in Fig.1 mean? They don't seem to match with the equations. d) What is the spatial resolution? e) Make unit naming consistent (e.g. Kg C m-2 day-1 vs. kg C m-2 day-1, p.5 l.10 vs. l.15) f) I suggest to rename 'Carbon concentration and fluxes' to 'Validation of carbon concentration and fluxes'

---

## Referee Comment (RC2) · Anonymous Referee #2 · 10 Oct 2017

**Representation of dissolved organic carbon in the JULES land surface model (vn4.4_JULES-DOCM) by Mahdi Nakhavali et al.**

The authors represent a model which calculates the DOC concentration to inland waters. They extended the JULES model for DOC, including soil carbon processes and leaching. This manuscript is a step towards a carbon model for aquatic systems and their export to the oceans.
In general, I think that the manuscript is well structured, but the description of the model needs some improvement. I recommend publishing the manuscript in GMD after revision.

Before I start with my comments, I must point out that I am giving feedback from a modelling point of view. My work on the global aquatic C cycle has just started, but I have a lot of expertise on global modelling.
From that point of view, I was very happy that both the abstract and the introduction start with sentences about global transport to the oceans. The importance of lateral transport is emphasized, but the model description itself does not contain a word on lateral transport. The model is actually a 1-D model and the outcome could be used to transport in the river network.

The second remark is the mentioning of the C cycle in the abstract "A model that represents the whole continuum from atmosphere to land and into the ocean would provide better understanding of the Earth's C cycle and hence more reliable historical or future projection" and introduction "Hence we need to move towards a boundless C cycle model which accounts for lateral fluxes". Why did you choose, after emphasizing C (as in total C) transport, to represent DOC only instead of modelling other species like POC, SOC and DIC as well?

Comments/Questions
Abstract line 29-30: I think that part of the leaching to the riverine system is explained by this model. The flux from groundwater or other sources going to the riverine network are not explained by this model. You could shortly elaborate on the relevance or importance of groundwater and give a short explanation on why you ignore it for now. The model comparison is done in the soil and not in the river network.
Introduction: Should be more clear on the objective/aim of the model study. Same for abstract.
Page 3, line 12: Why 3 meters deep?
Page 3, line 24: 9 PFTs at global scale. What about crops? They are mentioned in Figure S1. Names and the number of PFT do not match with Figure S1.
Page 4, eq 2: I think dz should be without subscript (2x). I don't see why it is important to calculate x? Remove eq 2?
Page 4, eq 3: I think z=1 and z=4 should be replaced by i=1 and i=4. This 1 and 4 is not explained yet (I think they are the four soil layers that will be used).
Page 4, line 30-31: These lines do not say anything. Which measurements? When and where taken? Why this remark here? The DOC is not mentioned here. Why are there continuous lines for measurements? For the modelled results? Eq. 3 only gives four outcomes…..
Page 5, line 3: In figure 1 I see four carbon pools added (two for lock and two for free).
Page 5, eq 4: k is indicator for labile or recalcitrant. But none of the other parameters is dependent on k, so why is k included?
Page 5, line 15: add i subscript in $F_S(S)$ and $F_T(T\_soil)$
Page 5, line 17: RothC formulations. Reference needed.
Page 5, eq 5: What is the unit of silt and clay?

Page 5, eq 6,7,8,9: What is S_CARB,DPM? Why twice substracting R_DPM? What is F_DOC,DPM? Please make the parameter names consistent. This system is solved for each soil layer, so why "i" is not in the equations? These formulas are not clear.

Page 5, eq 8,9: I don't understand why part of respiration (R_s) is flowing to BIO and HUM? Can you take another parameter name for beta_R? Confusing. What is F_BIO,IN?

Page 6, line 3: R_s neglects R_DOC but it is called total respiration?

Page 6, line 12: add i subscript in S_DOC and k subscript in K_DOC

Page 6, line 13: add i subscript in F_T(T_soil). Is this the same parameter as mentioned on the previous page?

Page 6, eq 11 and 12: Should there not be a sum over k (labile and recalcitrant) in these formulas?

Page 6, line 24-26: "The assumption … (k)." I don't know what you trying to say here….

Page 6, eq 14: I don't understand. The size of the labile DOC pool is the old value minus a flux plus total size of the adsorbed pool??

Page 7, eq 15: I don't understand. This means that size of adsorbed pool is equal to F_AD_i???

Page 7, line 4: add i subscript teta_v.

Page 7, line 7: add k,i subscript C_DOC.

Page 7, line 8: do you mean distance between midpoints of the soil layers?

Page 7, eq 16: add i subscript in the formula (C_DOC and z). Do I miss which direction the diffusion goes. Does it always go from layer 2 to 1 or layer 2 to layer 3? Then there should be a subscript i,j or something….

Page 7, eq 17 and 18: add subscript k and i to the formulas and in the text.

Page 7, eq 17 an 18: It is confusing to have another teta with another unit in these formulas.

Page 7, eq 19: I should expect that F_P is negative? Add all the k and i subscripts to this equation.

Page 7, line 29: What do you mean by main DOC model parameters? In what sense?

Page 8, line 32: explain Al/Ap, A/E and Cg

Page 9, line 29: analytical spin-up? What does that mean? Why the assumption that it must be a steady state?

Page 10, line 1: HWSD global data. Reference needed.

Page 10, line 6: "test the sensitivity of DOC related model parameters" On what? DOC leaching?

Page 10, line 6-7: Why are these parameters chosen? These parameters can say something about the inner-sensitivity of the model. But how about the inputs like for example assumptions on PFT or precipitation, temperature, and so on? What about choosing different number of soil layers?

Page 10, line 6: How can you change beta_z? It is calculated in eq. 3. But that is a normalization?? Should you not change z_0? And what are you changing? Beta_z for each layer?

Page 10, line 8: Remark. The method of changing one parameter at the time. This is a popular method. However, it renders no information on the effects of interactions of the parameters and that it covers only a limited part of the entire parameter space.

Page 10, line 8: Why 50%? 10 or 5% was also enough to say something about the sensitivity around the default values.

Page 20: Figure 1 is confusing. All 8 boxes are defined for all the four soil layers, but the diffusion and soil depth give the impression that for example DOC_lock_labile only are defined in the deeper soils. Suggestion: split up the figure with the 8 pools (left hand side) and the righthand side the diffusion. Leaching and soil depth and 4 boxes (which are of the lefthand side type).

Table 2 should be updated with the right subscripts and so on. "i" has not the unit m….

Table S1: Z_0 should be with lower case characters.

Table S4: What are the numbers in the matrix? Leaching? Unit?

General

Page 15, line 7: "Hence, it is important to introduce a depth-dependence decay rate for these parameters.". Now the sensitivity is used to draw this conclusion. But why not show the contribution of the DOC leaching of each soil layer? That should give a clear picture. I miss a kind of analysis of the importance of the different processes. Because the model that is proposed here has a lot of parameters. Is it possible to reduce the number of assumptions/soil layers/flows between the different pools? A broader sensitivity analysis could help.

What is the used spatial distribution? And the temporal resolution of the input?

---

## Author Comment (AC1) · 31 Oct 2017

Reviewer #1:

The authors represent a modelling development that makes it possible to have a more complete look at the global carbon cycle. They combine the well-established JULES model with a newly developed model for DOC, including soil carbon processes and leaching. This manuscript is therefore an important step towards a full carbon cycle understanding.
In general I think that the manuscript is well structured and the figures are helpful to understand the outcomes. However, there are some changes needed to make it more convincing. While I see some issues that should be clarified/solved first, I recommend publishing the manuscript in GMD after revision.

Thank you very much for your careful comments. We improved the manuscript following the reviewer's suggestions. Details are given below.

1. Please explain why the authors did not include the production of OC in soils and rivers (i.e. aquatic photosynthesis)?

Let us clarify what the model does and doesn't include.
The developments presented are made on the JULES, the Joint UK Land-Environment Simulator. The standard version of JULES, described in Clark et al. (2011), simulates vegetation and soil water and carbon processes, including the production of soil organic carbon following plant mortality, litter decomposition, respiration etc.
Our manuscript describes new developments in JULES in order to represent DOC cycling in soils, including DOC production is soils, DOC decomposition and respiration, as well as leaching of DOC from the soil to the aquatic environments.

However; the reviewer is right to say that we do not include fate of OC in rivers, this is beyond the scope of this work, as it would need to represent the full biogeochemistry of DOC in rivers (biological activity, sedimentation, gas transfer, etc.). Our developments lay the corner stone for a future representation of C cycling along the land-to-ocean continuum, including transport and transformation in the river network, as highlighted in our introduction and outlook. This is clarified now in the introduction.

2. Explain the additional value of adding Turkey Point and Guandaushi. To me it seems that the data from these two sides do not add much information, due to their much shorter time coverage. Also in the discussion, it is stated that e.g. Turkey Point is not really useful because it's located at a site that was agriculturally used until 1989. Possibly it would help to remove the Level 2-sites.

There is always a balance to find between very few sites that have well documented long-term measurements, and more sites, with shorter record and/or more complex history. In this work, we found that focusing on only 3 sites, all from Western European forests, would not suffice to gain confidence in our model, hence we decided to also include Turkey Point-89 and Guandaushi sites that we define as "level-2 sites", in order to explore broader climate/ecosystems domains. Turkey Point-89 provides a high productive forest ecosystem on a previously agricultural land. This is a typical system in North America where marginal agricultural land which have been afforested in recent decades. Also, we are keen to keep Turkey point as it is the only site that provides DOC measurements down to 100 cm, providing additional constraints on our model.
We find that our model has some difficulties in capturing dynamics at Turkey Point site. However, while we overestimate the total DOC in Turkey Point, we are able to reproduce the vertical profile of DOC in the soil.

3. Discuss the different methods in the study side description. Are there fundamental differences between 'suction cups' (p.8 l.22) and 'tension lysimeters' (p.8 l.32)? Are the observed values comparable?

First of all, we agree with the reviewer #1 that this is an important point which needs further clarification. For all sites, samples were collected using one of the two in situ soil solution extraction methods, namely suction cups or suction plates. We have, therefore, removed the term "lysimeter", which is often used incorrectly, and have replaced it with one of the two afore mentioned extraction methods. Where appropriate we have also included information on the material of the sampling device as well as the suction applied. This will allow the reader to better estimate the potential effects of different sampling methods on DOC concentrations obtained (Weihermüller et al. 2007). This is clarified in revised text:
For Hainich:
"At this site, soil solution samples were taken at three depths (5, 10 and 20 cm) using ceramic suction plates positioned at four different plots within the site. Samples were obtained by applying a tension of 100 hPa after each bi-weekly sampling occasion"

For Carlow:
"Samples were obtained by applying a tension of 400 hPa after each bi-weekly sampling occasion (Walmsley 2009)."
For Brasschaat:
"DOC samples were collected at three horizons of Al/Ap, A/E and Cg (Soil Classification Working Group 1998) referred to 10,35 and 75cm depth, by means of ceramic suction cups on a biweekly interval. Two days prior to sample collection a tension of 600 hPa was applied to each suction cup. Samples were collected at three locations and pooled into one composite sample per layer for analysis (Gielen et al. 2011)"
For Turkey Point-89:
"DOC sampling was attempted in monthly intervals at three depths of 25, 50 and 100 cm by means of porous suction cups..."
For Gunadaushi:
"DOC samples were collected at three depths of 15, 30 and 60 cm in three locations at bi-weekly interval by means of ceramic suction cups"

As samples were obtained by means of suction from a conical or plate-like porous sampling device we consider the differences due to methodological issues to be small compared to other soil solution collection methods such as passive samplers (pan lysimeters, wick samplers, resin boxes) and in particular extraction methods (centrifugation, chemical extractants such as KCl or CaCl2). The main difference between suction cups and suction plates is that the latter exhibit a larger sampling area and due to the 2D surface of the plates, the origin of the sampled solution is better defined which is important for mass balance estimations. It has also been shown that ceramic devices can absorb dissolved organic matter. As a result, concentrations in all but Carlow (where more suitable non-absorbing porous glass suction cups were used) may be underestimated. In addition, the suction applied will have an effect on the source of the soil solution collected (macro-, meso-, micropores). We are unable to quantify these effects in the current study. However, as mentioned above, we believe that the effect of different sampling methods on DOC concentrations are relatively small compared to potential differences due to sampling effects. Finally, as we are comparing observed values with the model simulation of Soil DOC concentration (not comparing observed values with each other), the difference in the sampling methods should not have a significant impact on the comparison of the model vs observed values.

When did this sampling take place (e.g. Turkey Point – 'samples could only be retrieved for 5 separate days'; all in summer? or winter?) (see also comment above (2.))

In Turkey point-89, due to the highly drained soil because of its sandy and dry condition, the sampling was taking place in spring and autumn when the soil was wet enough. Hence the dates of sampling were: 29-Nov-2004, 3-May-2005, 16-Jun-2005, 29-Jun-2005, 14-Oct-2005 (for more detail please see (Peichl et al. 2007)).
This is clarified in the revised manuscript:
"however, due to the dry sandy soils, samples could only be retrieved for 5 separate days of sampling after heavy rain fall events on 29-Nov-2004, 3-May-2005, 16-Jun-2005, 29-Jun-2005, 14-Oct-2005 (Peichl et al. 2007)."

4. Elaborate on the model input in more detail:
a) Have the data of FLUXNET and WATCH been somewhat corrected to be comparable?

No. These two data are not comparable. The FLUXNET data are the site-specific observed data. Hence, whenever we had access to the site measurements, we used them. The only site which we had to use WATCH data was Guandaushi, where we could not find any on-site measurements. Both forcing data were checked for any missing data and it was gap filled by linear interpolation. This is clarified in the revised manuscript:
"The FLUXNET database provides on-site meteorological data for each site that could be used as forcing for simulations in JULES, However, we had to use the global WATCH dataset (Weedon et al. 2010) as forcing for Guandaushi site where no on-site data was available. However, both forcing data were checked for any missing data and it was gap filled by linear interpolation."

b) Mention the names for the parameters as it's used in the tables and equations consistently (e.g. p.9 l.28: bulk density and clay content)

Corrected.

5. Please adapt the figures in a way that makes them easier to understand.
a) Fig.2 The extent can be smaller to better see where the sites are.

Corrected.

b) Fig.3 Decrease the y-axis range. A maximum of about 20 should be sufficient and the differences a better to see.

Corrected.

c) Fig .4 Decrease the y-axis range to a maximum of 40.

Corrected.

6. Combine both parts of the discussion to one. This would avoid the repetitions and can clearly combine all information/discussion on each of the sites.

Done as suggested, thank you.

Minor comments:

a) Define SOC at its first occurrence (p-2 l.28)

SOC first occurrence was defined in p-2 l.8.

b) Please use consistent and not confusing naming for the variables/parameters in the equations (e.g. R can mean 'run-off' and 'respiration')

Corrected. Runoff modified to Roff instead of R.

c) Connecting the equations to the arrows in the flow chart (Fig.1) would help in understanding the calculations. What do the numbers (1) to (12) in Fig.1 mean? They don't seem to match with the equations.

Sorry, this was unclear. Numbers were corresponding to process not to equations. As all arrows are defined within the text at the end of each process described (i.e. p.5 l.10: "($F_P$; arrows 1-4 Fig. 1)" indicating the process linked to the arrows in the model figure.)
In order to avoid potential confusion, we replaced the numbers in fig.1 with letters.

d) What is the spatial resolution?

The evaluation of model was performed on plot-scale, using climate forcing data, soil and land cover for one specific point, so no horizontal spatial dimension was involved. The model is however capable of running at each spatial resolution for which forcing data are available.
This is clarified in the revised manuscript  (p.10, l.15):
"The evaluation of model was performed on plot-scale, using climate forcing data, soil and land cover consistent with the site, no horizontal spatial dimension was involved."

e) Make unit naming consistent (e.g. Kg C m-2 day-1 vs. kg C m-2 day-1, p.5 l.10 vs. l.15)

Agreed. We now use "kg" throughout the text.

f) I suggest to rename 'Carbon concentration and fluxes' to 'Validation of carbon concentration and fluxes'

Corrected.

---

## Author Comment (AC2) · 31 Oct 2017

Reviewer #2:

The authors represent a model which calculates the DOC concentration to inland waters. They extended the JULES model for DOC, including soil carbon processes and leaching. This manuscript is a step towards a carbon model for aquatic systems and their export to the oceans.
In general, I think that the manuscript is well structured, but the description of the model needs some improvement. I recommend publishing the manuscript in GMD after revision.

Thank you very much for your careful comments. We improved the manuscript following the reviewer's suggestions. Details are given below.

Before I start with my comments, I must point out that I am giving feedback from a modelling point of view. My work on the global aquatic C cycle has just started, but I have a lot of expertise on global modelling.
From that point of view, I was very happy that both the abstract and the introduction start with sentences about global transport to the oceans. The importance of lateral transport is emphasized, but the model description itself does not contain a word on lateral transport. The model is actually a 1-D model and the outcome could be used to transport in the river network.
The second remark is the mentioning of the C cycle in the abstract "A model that represents the whole continuum from atmosphere to land and into the ocean would provide better understanding of the Earth's C cycle and hence more reliable historical or future projection" and introduction "Hence we need to move towards a boundless C cycle model which accounts for lateral fluxes". Why did you choose, after emphasizing C (as in total C) transport, to represent DOC only instead of modelling other species like POC, SOC and DIC as well?

We emphasize the importance of lateral transfers and C cycling along the land-to-ocean aquatic continuum in the abstract and the introduction, as this marks the ultimate goal of our model developments. Nevertheless, our manuscript represents just a first step in that development. At later stages of the overall model development, other processes and C species will be dealt with as well. This is now clarified in the abstract:

"A first and critical step in that direction is to include processes representing production and export of dissolved organic carbon in soils. Here we present an original representation of Dissolved Organic C (DOC) processes in the Joint UK Land Environment Simulator (JULES-DOCM) that integrates a representation of DOC production in terrestrial ecosystems based on incomplete decomposition of organic matter, DOC decomposition within the soil column, and DOC export to the river network via leaching."

As for representing DOC only, instead of POC, SOC and DIC, the reviewer is right, that ideally, we should represent carbon exports in all forms.
According to Meybeck (1982), DOC exports to the coast represent about 37% of C taken up on land from the atmosphere and being laterally exported along the river network. DIC is also a large source of carbon to rivers (potentially larger than DOC), but DIC sources are driven by very different processes such as rock erosion, that are not directly connected to soil organic carbon and the terrestrial carbon cycle. As a first priority, we then decided to focus the JULES developments on DOC processes. Different forms of C will need different processes to be represented in future steps of implementing the land-to-ocean aquatic continuum into the representation of the global C cycle. For instance, the simulation of POC transports would require the representation of erosion, sediment transport and autotrophic production. The representation of DIC would require the representation of weathering processes and water-air gas exchanges. This is now clarified in the introduction:

"Other forms of C need different processes to be represented to fully represent the land-to-ocean aquatic continuum of the global C cycle. Hence future work should include DIC and POC export from soils as well as the fate of all exported carbon in the river system.''

Comments/Questions

Abstract line 29-30: I think that part of the leaching to the riverine system is explained by this model. The flux from groundwater or other sources going to the riverine network are not explained by this model. You could shortly elaborate on the relevance or importance of groundwater and give a short explanation on why you ignore it for now. The model comparison is done in the soil and not in the river network.

As in most global land surface models, a ground water aquifer is not directly represented in JULES-DOCM. Runoff from soils is simply represented as two components, a surface runoff and a subsurface runoff. The subsurface runoff includes the drainage from the bottom of the 3m soil column, and thus somehow mimics the ground water base flow, in terms of water as well as in terms of DOC exports. This information is now added to the leaching section in model (p.7, line16-17):
"However subsurface runoff is also representing the drainage from the bottom of the 3m soil column, and thus mimics the groundwater base flow, in terms of water as well as in terms of DOC exports"

Please, also note that in this manuscript, we focus on DOC cycling within the soil column, and we do not yet represent C fluxes in the rivers. For that reason, we compare our simulation results against observed DOC concentrations in the soil solution. Carbon fluxes in rivers are in addition affected by decomposition of DOC, additional sources of DOC from the decomposition of POC and the evasion of $CO_2$ to the atmosphere. Comparing the simulated leaching of DOC to the river against observed DOC concentrations at some downstream sampling location would not be valid because of the non-conservative behaviour of DOC in the river.

Introduction: Should be more clear on the objective/aim of the model study. Same for abstract.

We agree with reviewer #2 and revised abstract and introduction accordingly.

Page 3, line 12: Why 3 meters deep?

JULES default soil depth is set at 3 meters. The root profile, the soil C stocks and the soil hydrology are all simulated over that 3-meter soil profile, which we used here for the representation of DOC. Moreover, soil depth was not always available at measurement sites. Therefore, we decided to keep the default values for which the JULES model was developed. This information is now added to revised text (p3, l.11):
"The aim of this study is to include a representation of DOC produced in terrestrial soils down to 3 meters (as soil hydrology and Carbon are simulated over 3 meter soil profile in JULES)"

Page 3, line 24: 9 PFTs at global scale. What about crops? They are mentioned in Figure S1.
Names and the number of PFT do not match with Figure S1.

We thank the reviewer for this comment. Table S1 is giving Z0 for the PFTs as described in Jobbágy & Jackson data, not the JULES PFTs, sorry for the confusion. We added another table (Table. S2) giving Z0 for the JULES PFTs.

As for crops, in this version of JULES crops are classified as C3 and C4 grasslands. Note that there is a separate version of JULES with improved representation of crops (Osborne et al., 2015), not used here as our main focus is on natural ecosystems

Page 4, eq 2: I think dz should be without subscript (2x). I don't see why it is important to calculate x? Remove eq 2?

We corrected the notation of dz.
Please note that $x$ is the ration of SOC content within the first 1 meter of soil relative to the 3-meter profile for different biomes as given by Jobbágy & Jackson (in their Table. 3) (Jobbágy & Jackson 2000) which is used to extrapolate a profile of soil C concentrations. This is clarified in the revised manuscript.

Page 4, eq 3: I think z=1 and z=4 should be replaced by i=1 and i=4. This 1 and 4 is not explained yet (I think they are the four soil layers that will be used).

Corrected. We replaced z with i. This is indicating the normalized weighting factors for all four soil layers (i).

Page 4, line 30-31: These lines do not say anything. Which measurements? When and where taken? Why this remark here? The DOC is not mentioned here. Why are there continuous lines for measurements? For the modelled results? Eq. 3 only gives four outcomes…..

We largely rewrote section 2.2.1, clarifying the approach to distribute organic carbon (calculated as a bulk stock) in the vertical to serve as input for the DOC model.

Page 5, line 3: In figure 1 I see four carbon pools added (two for lock and two for free).

Corrected.

Page 5, eq 4: k is indicator for labile or recalcitrant. But none of the other parameters is dependent on k, so why is k included?

We thank reviewer #2 comment on this equation. We added a subscript k for the soil C stocks Sc (now $Sc_k$), as the soil carbon pool defines which amounts of DOC produced go to labile and refractory DOC, respectively.

Page 5, line 15: add i subscript in F_S(S) and F_T(T_soil)

Corrected.

Page 5, line 17: RothC formulations. Reference needed.

Added.

Page 5, eq 5: What is the unit of silt and clay?

Fraction. Now it is added. We also changed the values from % to fraction in table 4.

Page 5, eq 6,7,8,9: What is S_CARB,DPM? Why twice substracting R_DPM? What is F_DOC,DPM? Please make the parameter names consistent. This system is solved for each soil layer, so why "i" is not in the equations? These formulas are not clear.

We thank reviewer #2 comment on these equations.

There was some typo in Equation5. This is corrected now. Variables names have also been checked and made consistent. We also corrected the equations adding "i" to the updates of pools based on the sum of DOC processes in all layers.

Page 5, eq 8,9: I don't understand why part of respiration (R_s) is flowing to BIO and HUM?
Can you take another parameter name for beta_R? Confusing. What is F_BIO,IN?

In RothC model the assumption is that part of decomposed carbon (B_r) is released to the atmosphere and the remaining fractrion (1- B_r) is feeding microorganism in biomass (BIO) pool or is stored in the soil as the recalcitrant form, humus (HUM) with a slowest decomposition rate. These terms are fully described in JULES description model (Clark et al. 2011).
These are clarified in the revised manuscript:
"where in RothC model fraction ($f_{DPM}$) of litter fall ($\Lambda_c$) is directed to DPM and RPM depending on vegetation type. C pools are subjected to decomposition. Part of decomposed C as a fraction ($1-B_R$) of total respiration ($R_s = R_{DPM}+ R_{RPM}+ R_{BIO}+ R_{HUM}+ R_{DOC}$) is partially feeding microorganisms in soil (BIO) and partially stored as recalcitrant C in soil (HUM) depending on soil texture and the rest ($B_R$) is released to the atmosphere."

We changed beta_R to B_r, to avoid confusion with the beta we use in equation 3.

F BIO,IN is CUE fraction of decomposed DOC which is going back to biomass pool (described in eq 11)

Page 6, line 3: R_s neglects R_DOC but it is called total respiration?

R_s in code is indeed the total respiration including the R_DOC it in code. This was a typo that we corrected to:
R_S = R_DPM+R_RPM+R_BIO+R_HUM+R_DOC

Page 6, line 12: add i subscript in S_DOC and k subscript in K_DOC

Corrected.

Page 6, line 13: add i subscript in F_T(T_soil). Is this the same parameter as mentioned on the previous page?

Corrected. Yes, it is the same parameter. This is added to revised text:

"$F_T(T_{soil})_i$ is the soil temperature rate modifier within each soil layer (i) same as in eq.4"

Page 6, eq 11 and 12: Should there not be a sum over k (labile and recalcitrant) in these formulas?

Corrected. We added the sum sign indicating that at the end the BIO_IN flux will be the sum of both labile and recalcitrant decomposed DOC.

Page 6, line 24-26: "The assumption … (k)." I don't know what you trying to say here….
Page 6, eq 14: I don't understand. The size of the labile DOC pool is the old value minus a flux plus total size of the adsorbed pool??
Page 7, eq 15: I don't understand. This means that size of adsorbed pool is equal to F_AD_i???

We thank reviewer #2 for the comment on the adsorption/desorption.
We revised the manuscript:

"For adsorption/desorption, a constant sorption equilibrium distribution coefficient ($K_D$) is used to partition DOC in dissolved and adsorbed phases. The assumption is that DOC in the labile or recalcitrant pool is proportionally distributed between adsorbed DOC ($S_{DOCad}$) and dissolved DOC pools ($S_{DOC}$ in soluble phase) depending on $K_D$ from each soil layer(i) and DOC pool (k). Hence if the potentially adsorbed DOC fraction ($AD\_pot_i$) compared to the size of the actually adsorbed DOC ($S_{DOC_{ad\,k,i}}$) is positive then this fraction will be adsorbed and added to the adsorbed DOC pool, and if it is negative then this fraction will be desorbed and added to dissolved DOC pool per model time step.

These terms for DOC labile and recalcitrant pools in JULES-DOCM are as follow (arrow: i and j, Fig. 1):

$$AD\_pot_i = S_{DOC_{k,i}} \times K_D \times \frac{BK}{\theta v_i} \tag{eq.13}$$

$$S_{DOC_{k,i}} = S_{DOC_{k,i}} - \left( AD\_pot_i - S_{DOC_{ad\,k,i}} \right) \tag{eq.14}$$

$$S_{DOC_{ad\,k,i}} = S_{DOC_{ad\,k,i}} + \left( AD\_pot_i - S_{DOC_{ad\,k,i}} \right) \tag{eq.15}"$$

Also in order to make it easier to read we replaced "locked DOC" with "adsorbed DOC" and "free DOC" with "dissolved DOC".

Page 7, line 4: add i subscript teta_v.

Corrected.

Page 7, line 7: add k,i subscript C_DOC.

Corrected.

Page 7, line 8: do you mean distance between midpoints of the soil layers?

Yes. We replaced "the distance (z_i) between every two soil depths" to "distance (z_i) between midpoints of the soil layers"

Page 7, eq 16: add i subscript in the formula (C_DOC and z).

Corrected subscript.

Do I miss which direction the diffusion goes. Does it always go from layer 2 to 1 or layer 2 to layer 3? Then there should be a subscript i,j or something….

Agreed. We changed the subscript to: subscript i for downward flow, and j for upward flow of diffusion.

Page 7, eq 17 and 18: add subscript k and i to the formulas and in the text.

Corrected. We also added that top soil is the sum of first and second soil layer, and bottom soil is sum of thirds and fourth soil layer.

Page 7, eq 17 an 18: It is confusing to have another teta with another unit in these formulas.

We changed teta_s with T_s.

Page 7, eq 19: I should expect that F_P is negative?

F_P is the production of DOC, it is never below 0.

Add all the k and i subscripts to this equation.

Corrected.

Page 7, line 29: What do you mean by main DOC model parameters? In what sense?

Default model parameters. We changed "main" to "default".

Page 8, line 32: explain Al/Ap, A/E and Cg

Reference added for soil horizons.

Page 9, line 29: analytical spin-up? What does that mean? Why the assumption that it must be a steady state?

We removed the terms "analytical spin-up". In order to have the present-day C, we did the spin-up looping 300 times over each site until we reached the steady state for C in soil.  This is revised in the manuscript now.
"The model was first spun-up looping over period 1996 to 2014 until all the soil variables reached a steady state."

Page 10, line 1: HWSD global data. Reference needed.

Added.

Page 10, line 6: "test the sensitivity of DOC related model parameters" On what? DOC leaching?

Sorry, this was unclear. We tested them on the DOC concentration in different depths of the soil profile. We added this information in the revised manuscript:

"In order to test the sensitivity of DOC related model parameters on the DOC concentration in different depths of the soil profile, simulations were performed with varying values for $z_0$, $\tau_z$ and DOC controlling parameters such as $K_{DOC(labile)}$, $K_{DOC\ (recalcitrant)}$, $D_f$, $CUE$, $K_D$ and $D$ (Table 1)."

Page 10, line 6-7: Why are these parameters chosen? These parameters can say something about the inner-sensitivity of the model. But how about the inputs like for example assumptions on PFT or precipitation, temperature, and so on? What about choosing different number of soil layers?

We ran the sensitivity analyses on the rate constants which we took from the literature and which could be subject to a recalibration. Simulation results may as well be sensitive to forcing data used, but that is not the point of this model development study where we used on-site observations of climate instead of global forcing data which would be subject to more uncertainties. In JULES-DOCM, the soil profile depth and number of layers is fixed and cannot be changed, because of the dependence on the representation of soil hydrology.

Page 10, line 6: How can you change beta_z? It is calculated in eq. 3. But that is a normalization?? Should you not change z_0? And what are you changing? Beta_z for each layer?

Reviewer #2 is absolutely right. We did indeed changed z_0 and based on that got the new normalized beta_z for each layer. We clarified this in the revised text

Page 10, line 8: Remark. The method of changing one parameter at the time. This is a popular method. However, it renders no information on the effects of interactions of the parameters and that it covers only a limited part of the entire parameter space.

We agree with reviewer #2 that there could be some interactions between sensitivity of different parameters, but testing this was beyond the scope of this study.

Page 10, line 8: Why 50%? 10 or 5% was also enough to say something about the sensitivity around the default values.

Since the derived model parameters from literature already had their own level of uncertainty, for instance CUE which has more than 50% or $K_{DOC}$ with 5-40% of uncertainty, we took the 50% of change to test all the parameters at the reasonable degree.

Page 20: Figure 1 is confusing. All 8 boxes are defined for all the four soil layers, but the diffusion and soil depth give the impression that for example DOC_lock_labile only are defined in the deeper soils. Suggestion: split up the figure with the 8 pools (left hand side) and the righthand side the diffusion. Leaching and soil depth and 4 boxes (which are of the lefthand side type).

Corrected as suggested.

Table 2 should be updated with the right subscripts and so on. "i" has not the unit m….

"i" is corrected.
Subscripts are defined within text. In Table 2, we omitted subscripts for reasons of readability.

Table S1: Z_0 should be with lower case characters

Corrected.

Table S4: What are the numbers in the matrix? Leaching? Unit?

Table S4 lists the values for Figure 8 (Relative change in simulated DOC (%) for a +50% and -50% changes in model parameters). We added this now to Table S4 description.

**General**

Page 15, line 7: "Hence, it is important to introduce a depth-dependence decay rate for these parameters.". Now the sensitivity is used to draw this conclusion. But why not show the contribution of the DOC leaching of each soil layer? That should give a clear picture. I miss a kind of analysis of the importance of the different processes. Because the model that is proposed here has a lot of parameters. Is it possible to reduce the number of assumptions/soil layers/flows between the different pools? A broader sensitivity analysis could help.

Please note that we are not representing the DOC leaching from each soil layer. As we described in leaching of model, the first and second layer together are considered as the top soil and leaching is taken from it. The update

of these two layers will be based on the proportion of leached DOC compared to the DOC concentration in each of these layers. The same applied to the third and fourth layer as the sub soil leaching.

We are representing only the key processes for DOC including production, decomposition, adsorption/desorption, diffusion and leaching. Hence, we do not feel we could significantly reduce our model assumptions here. That being said, we find that adsorption/desorption was not making any significant change to our simulation and is probably of second order in the estimate of DOC soil concentration and export. We revised the sensitivity section in discussion:

"The sensitivity tests indicate that the parameters controlling SOC concentrations in the soil profile ($Z_0$ and $\tau_z$) and the recalcitrant DOC residence time ($K_{DOC\ (recalcitrant)}$) have the most significant effect on soil DOC concentration, which indicates the importance of factors controlling DOC sources."

Regarding the change of soil layers/flows please look at answer to your comment on page 10, line 6-7.

What is the used spatial distribution? And the temporal resolution of the input?

As mentioned in our response to reviewer #1, the evaluation of model was performed at plot-scale using 1 dimensional climate forcing, thus no spatial resolution. Temporal resolution of the input is 30 minutes. This is now added to manuscript (p.10, l.6):

"The meteorological forcing is provided at the measurement site level (no explicit spatial resolution) and includes the downward shortwave and longwave radiation at the surface (W m-2), rainfall (kg m-2 s-1), snowfall (kg m-2 s-1), wind speed (m s-1), atmospheric temperature (K), atmospheric specific humidity (kg kg-1) and air pressure at the surface (Pa) at half an hour time step"

---

## Author Response (AR2)

First of all, I am happy with the all changes that are made after the first revision. It improved the article! However I still have some remarks.

Thank you very much for your careful comments. We improved the manuscript following the reviewer's suggestions. Details are given below.

First I don't understand the reaction of the authors on reviewer #2:
"Page 10, line 8: Remark. The method of changing one parameter at the time. This is a popular method. However, it renders no information on the effects of interactions of the parameters and that it covers only a limited part of the entire parameter space.
We agree with reviewer #2 that there could be some interactions between sensitivity of different parameters, but testing this was beyond the scope of this study."
My question is: why is performing a good sensitivity analyses beyond the scope of this study? This is a weak reply of the authors on this remark.

Sorry, we failed to clearly explain the challenge of doing what the reviewer is asking. We explored the sensitivity of our model results at three different sites for 8 parameters. Taking a high (+50%) and low (-50+%) value for each parameter individually leads to 16 simulations per site, i;e. 48 simulations in total. As changing a parameter's value has a potential impact on the whole carbon dynamic, each simulation required a new spin-up, looping 300 time over the climate forcing.
A full factorial experiment as suggested by the reviewer would need to change up to 8 parameters at the same time. For 8 parameters with 3 possible values each, the number of combination is $3^8$, that is 6561 simulations per site, i.e. almost 20,000 simulations in total.
This is well beyond the capability of our computation resources, and hence is not feasible in the framework of this study. Hence we chose to stick with the traditional simple method of changing one parameter at a time.
However, the effect of "interactions of parameters" is a valid point, and in the final manuscript we mention that limitation of our sensitivity analysis:
"We note that our sensitivity analysis, changing one parameter at a time, does not investigate the potential interactions between different parameters."

My second remark is on the following reaction:
"Page 10, line 8: Why 50%? 10 or 5% was also enough to say something about the sensitivity around the default values.
Since the derived model parameters from literature already had their own level of uncertainty, for instance CUE which has more than 50% or K DOC with 5-40% of uncertainty, we took the 50% of change to test all the parameters at the reasonable degree."
Firstly I got the impression that the authors are a little bit confused. An uncertainty range is not needed to perform a sensitivity analyses. To avoid effects of a possible non-linear behaviour of the model, a small range is in general better. Besides this, the authors claim that they took 50% change for all parameters. However table S4 says something different. Also table 1 says something different. For example parameter D_f has a change of 33%. This table (and also table 2) has also a unit of Kg, which was one of the remarks of reviewer #1.

From the results in figure 8, we show that even with a change in parameter values of +/- 50%, for most parameters, the results hardly change.  A 5-10% change in those parameters would lead to virtually no changes in the simulated DOC.  Also, our model is essentially linear (see main model's equations in the manuscript, most fluxes follows first order reaction rate), so we are not too worried about larger perturbations. We still believe that changing the parameter values by +/- 50% is a valid choice to clearly show the insensitivity of the model to some parameters, while still exploring possible ranges for these parameters.

Indeed, we did not use the 50% range for the parameter D_f, as D_f depends on the soil type, with values ranging between 0.25 (100% clay and/or silt) and 1.0 (100% sand), for slope factor of 0.75 ( see equation 5). It would not make sense to have a slope factor larger than 1.0 in this equation, it would lead to non-physical negative values of D_f for 100% clay and/or silt. Hence our restricted range for that parameter, going from
0.5 (-33%), to 1.0 (+33%). This is now clarified in the final version of the MS:
"In order to test the sensitivity of DOC related model parameters on the DOC concentration in different depths of the soil profile, simulations were performed with varying values for $z_0$, $\tau_z$ and DOC controlling parameters such as $K_{DOC\ (labile)}$, $K_{DOC\ (recalcitrant)}$, slope parameter of $D_f$, CUE, $K_D$ and D (Table 1).
In total, 16 runs were performed by modifying each parameter once by increasing it 50% and once by decreasing it by 50%, except for the slope parameter controlling $D_f$ (eq. 5) which was changed by 33% to remain within the physical boundaries."

Kg is corrected to kg.

Other remarks:
Figure 6: Unit Kg is used.

Corrected

Figure 8: This is a representation of information that is given in table S4 and S5. I don't understand this information. Table S4 gives 16 results. I think that set-1 is used to calculate the 'relative change' (how defined?). The second set has a global beta_z, but that is not in the figure 8, nor in table S5. But why not change z_0 into two directions? I don't know. Why is z_0 for Carlow coloured red and having a positive relative change? Please make this information consistent with each other.

There was a typo in supplement document:
Beta_z is corrected to $z_0$.
As well as in S4 we changed Set-1 from default to PFT based $z_0$ and Set-2 from global beta_z to global $z_0$
.
Despite all the other sites, Carlow was the only site that using global value of $z_0$ was giving the positive change. This is now clarified in text:
"In contrast to the other sites, global $z_0$ leads to a low but still significant positive change of 6.5% in simulated DOC within 3 meters of soil (p-value <0.05, Table S6) (Fig. 8-c)."

Table S1 and S2: $Z_0$ instead of $z_0$.

Corrected

Figure S3: Unit Kg is used. Which depth is the DOC concentration?

Corrected. Down to 3 m averaged. This is added to revised text.

Figure S4: "-" sign is missing in the unit on the y axis

Corrected

pag 5 supplement: references to figure S4 and S5 are wrong. On pag 7 the references to tables are wrong. Please check all the references to tables and figures!

Corrected

Figure S5: "-" sign is missing in the unit on the y-axis and x-axis.

Corrected

Figure S6: C is missing in the unit on the y-axis.

Corrected

Is the reference at the end still needed?

removed

My advice is that the authors should go through the manuscript to fix the problems and to make the supplementary material consistent with the main text.

[revised manuscript text omitted]

Supporting information

Table S1. $z_0$ values for each PFT

| PFT | $z_0$ |
|---|---|
| Boreal forest | 0.775625 |
| Crops | 1.13717 |
| Deserts | 1.67113 |
| Sclerophyllous shrubs | 1.22839 |
| Temperate deciduous forest | 0.725914 |
| Temperate evergreen forest | 0.857235 |
| Temperate grassland | 1.22839 |
| Tropical deciduous forest | 1.67113 |
| Tropical evergreen forest | 1.0188 |
| Temperate grassland/savana | 1.45185 |
| Tundra | 0.656898 |

Table S2. $z_0$ values for each PFT in JULES-DOCM

| JULES PFT | $z_0$ |
|---|---|
| Tropical broadleaf evergreen forest | 1.0188 |
| Temperate broadleaf evergreen forest | 0.857235 |
| broadleaf deciduous forest | 0.725914 |
| needle-leaf evergreen forest | 0.857235 |
| needle-leaf deciduous forest | 0.725914 |
| C3 grass | 1.13717 |
| C4 grass | 1.13717 |
| Evergreen shrubs | 1.22839 |
| Deciduous shrubs | 1.22839 |

Table S3. Anova test results for Carbon fluxes (Df: Degree of freedom, Sum sq: sum of squares, Mean sq: mean of squares, Pr: p-value)

| | Df | Sum sq | Mean Sq | F value | Pr(>F) |
|---|---|---|---|---|---|
| **GPP** | | | | | |
| | | | | | |
| Hainich | 1 | 102955 | 102955 | 5.352 | 0.0343 |
| | 16 | 307785 | 19237 | | |
| | | | | | |
| Brasschaat | 1 | 280924 | 280924 | 62.27 | 1.33E-05 |
| | 10 | 45117 | 4512 | | |
| | | | | | |
| Turkey Point-89 | 1 | 9295 | 9295 | 4.714 | 0.162 |
| | 2 | 3943 | 1972 | | |
| **NPP** | | | | | |
| | | | | | |
| Hainich | 1 | 88254 | 88254 | 7.222 | 0.0362 |
| | 6 | 73323 | 12220 | | |
| Turkey Point-89 | 1 | 39632 | 39632 | 7.154 | 0.116 |
| | 2 | 11080 | 5530 | | |
| **SOIL RESPIRATION** | | | | | |
| | | | | | |
| Hainich | 1 | 1400 | 1400 | 0.06 | 0.815 |
| | 6 | 140896 | 23483 | | |
| | | | | | |
| Brasschaat | 1 | 160497 | 160947 | 77.44 | 1.40E-06 |
| | 12 | 24870 | 2073 | | |
| | | | | | |
| Turkey Point-89 | 1 | 98114 | 98114 | 9.687 | 0.0896 |
| | 2 | 20256 | 10128 | | |

ANOVA

We include all the sensitivity runs for Level-1 sites: Hainich, Brasschaat and Carlow for all the depths where the measurements were available. Red points are indicating measurements where black points are values from model (Fig. S2). Also representing Level-2 sites: Turkey Point 89 and Guandaushi comparison of modelled DOC versus measured in deeper soil depths (Fig. S3).

[Figure]

Figure S1. Sensitivity tests (black line) versus measurements (red dot) at a) 20cm depth – 2004-2013, Hainich b) 10cm depth – 2006-2010, Brasschaat c) 35cm depth – 2006-2010, Brasschaat d) 75cm depth – 2006-2010, Brasschaat e) 10 to28cm depth – 2006-2009, Carlow f) 28 to78cm depth – 2006-2009, Carlow; X axis is year and Y axis is DOC concentration in mg C L$^{-1}$. Parameter sets description and values in Table 1

[Figure]

Figure S2. DOC concentration (mg C L-1) for (a) Guandaushi at 30 cm measured (black dot: Chinese Fir, green dot: natural hardwood, orange dot: secondary wood, red square: mean, red line: standard deviation) and simulated (black lines) and for (b) Turkey Point 89 at 100 cm measured (red dots) and simulated (black lines, grey line: standard deviation).

We examine the hydrology of the model and its interaction with DOC concentration and leaching for Level-1 sites: Carlow and Brasschat (Fig. S3) and overall model performance in DOC representation in all depths by comparing modelled versus measurements during study period in Hainich, Brasschaat and Carlow (Fig. S4).

[Figure]

Figure S3. a) Precipitation, runoff, DOC leaching and DOC concentration (down to 3 m averaged) in Carlow from 2006 to 2009 b) in Brasschaat from 2006 to 2010 simulated data with JULES-DOCM

[Figure]

Figure S4. a) Measured vs modelled DOC (mg C L-1) with default set in Hainich from 2006 to 2013 at 10 and 20cm b) in Brasschaat from 2006 to 2010 at 10,35 and 75cm c) in Carlow from 2006 to 2009 at 10,10 to 28 and 28 to 78cm

Sensitivity of model parameters (Table S4) was tested in Level-1 Sites for the depths where the DOC measurements where available (Fig. S5) and the results were reported in percentage of change compared to default parameters set (Table S5). Anova test was used in order to determine each parameter's impact significance on DOC representation (Table S6).

Table S4. JULES-DOCM parameters set for sensitivity test

| ID | Description |
|---|---|
| SET-1 | PFT based $z_0$ |
| SET -2 | Global $z_0$ |
| SET -3 | +50% $\tau_z$ |
| SET -4 | -50% $\tau_z$ |
| SET -5 | +50% $K_{DOC,\ labile}$ |
| SET -6 | -50% $K_{DOC,\ labile}$ |
| SET -7 | +50% $K_{DOC,\ recalcitrant}$ |
| SET -8 | -50% $K_{DOC,\ recalcitrant}$ |
| SET -9 | +25% $D_f$ |
| SET -10 | -25% $D_f$ |
| SET -11 | +50% CUE |
| SET -12 | -50% CUE |
| SET -13 | +50% D |
| SET -14 | -50% D |
| SET -15 | +50% $K_D$ |
| SET -16 | -50% $K_D$ |

[Figure]

Figure S5. a) Measured (X axis) vs modelled (Y axis) DOC sensitivity runs (Table1) in Hainich from 2006 to 2013 at 10 and 20cm b) in Brasschaat from 2006 to 2010 at 10,35 and 75cm c) in Carlow from 2006 to 2009 at 10,10 to 28 and 28 to 78cm

Table S5. Relative change in simulated DOC (%) for a +50% and -50% changes in model parameters for Hainich, Carlow and Brasschaat

| HAINICH | $Z_0$ | $\tau_Z$ | $K_{DOC-LABILE}$ | $K_{DOC-RECALCITRANT}$ | $D_F$ | CUE | D | $K_D$ |
|---|---|---|---|---|---|---|---|---|
| 50% | - (PFT based z₀) | 24.82748 | 0.2746118 | 18.51532 | -0.590175 | -0.06117 | -0.04519973 | -0.05364935 |
| -50% | -25.12175 (global z₀) | - 36.45213 | -0.2729602 | -29.18424 | 0.586697 | 0.06068464 | 0.04545651 | 0.05483722 |
| **CARLOW** | | | | | | | | |
| 50% | - (PFT based z₀) | 16.81662 | 0.9795210 | 18.77268 | -0.2522954 | -0.08957044 | 0.1873639 | -0.05575369 |
| -50% | 6.52764 (global z₀) | -31.50205 | -0.9754175 | -27.40512 | 0.2517171 | 0.08983245 | -0.2774403 | 0.05659863 |
| **BRASSCHAAT** | | | | | | | | |
| 50% | - (PFT based z₀) | 27.52056 | 0.5294166 | 23.45682 | -0.1300973 | -0.1176923 | -0.3806475 | -1.256365 |
| -50% | -40.6144 (global z₀) | -38.92471 | -0.5120930 | -36.20752 | 0.1305755 | 0.1183834 | 0.3794148 | 1.571266 |

Table S6. Anova test results for sensitivity test of Level-1 sites

| DOC Hainich | Df | Sum sq | Mean Sq | F value | Pr(>F) |
|---|---|---|---|---|---|
| set-2 | 1 | 10686 | 10686 | 1066 | **2.00E-16** |
| set-3 | 1 | 10437 | 10437 | 491.9 | **2.00E-16** |
| set-4 | 1 | 22499 | 22499 | 2572 | **2.00E-16** |
| set-5 | 1 | 1 | 1.277 | 0.094 | 0.759 |
| set-6 | 1 | 1 | 1.267 | 0.093 | 0.76 |
| set-7 | 1 | 5805 | 5805 | 382.7 | **2.00E-16** |
| set-8 | 1 | 14422 | 14422 | 1322 | **2.00E-16** |
| set-9 | 1 | 6 | 5.898 | 0.437 | 0.508 |
| set10 | 1 | 6 | 5.828 | 0.427 | 0.514 |
| set11 | 1 | 0 | 0.063 | 0.005 | 0.946 |
| set12 | 1 | 0 | 0.062 | 0.005 | 0.946 |
| set13 | 1 | 0 | 0.035 | 0.003 | 0.96 |
| set14 | 1 | 0 | 0.035 | 0.003 | 0.96 |
| set15 | 1 | 0 | 0.049 | 0.004 | 0.952 |
| set16 | 1 | 0 | 0.049 | 0.004 | 0.952 |

| DOC Carlow | Df | Sum sq | Mean Sq | F value | Pr(>F) |
|---|---|---|---|---|---|
| set-2 | 1 | 116 | 115.98 | 20.27 | **6.81E-06** |
| set-3 | 1 | 770 | 769.7 | 109.73 | **2.00E-16** |
| set-4 | 1 | 2701 | 2701.1 | 703.7 | **2.00E-16** |
| set-5 | 1 | 3 | 2.611 | 0.486 | 0.486 |
| set-6 | 1 | 3 | 2.59 | 0.49 | 0.484 |
| set-7 | 1 | 959 | 959.2 | 145.6 | **2.00E-16** |
| set-8 | 1 | 2044 | 2044.2 | 488.3 | **2.00E-16** |
| set-9 | 1 | 0 | 0.173 | 0.033 | 0.857 |
| set10 | 1 | 0 | 0.172 | 0.032 | 0.857 |
| set11 | 1 | 0 | 0.022 | 0.004 | 0.949 |
| set12 | 1 | 0 | 0.022 | 0.004 | 0.949 |
| set13 | 1 | 0 | 0.096 | 0.018 | 0.894 |
| set14 | 1 | 0 | 0.096 | 0.018 | 0.894 |
| set15 | 1 | 0 | 0.008 | 0.002 | 0.968 |
| set16 | 1 | 0 | 0.008 | 0.002 | 0.968 |

| DOC Brasschaat | Df | Sum sq | Mean Sq | F value | Pr(>F) |
|---|---|---|---|---|---|
| set-2 | 1 | 98664 | 98664 | 5924 | **2.00E-16** |
| set-3 | 1 | 45301 | 45301 | 126.7 | **2.00E-16** |
| set-4 | 1 | 90625 | 90625 | 585.2 | **2.00E-16** |
| set-5 | 1 | 17 | 16.76 | 0.069 | 0.792 |
| set-6 | 1 | 16 | 15.69 | 0.065 | 0.798 |
| set-7 | 1 | 32911 | 32911 | 109.1 | **2.00E-16** |
| set-8 | 1 | 78414 | 78414 | 462.8 | **2.00E-16** |
| set-9 | 1 | 1 | 1.01 | 0.004 | 0.948 |
| set10 | 1 | 1 | 1.02 | 0.004 | 0.948 |
| set11 | 1 | 1 | 0.83 | 0.003 | 0.953 |
| set12 | 1 | 1 | 0.83 | 0.003 | 0.953 |
| set13 | 1 | 9 | 8.67 | 0.036 | 0.849 |
| set14 | 1 | 9 | 8.61 | 0.036 | 0.85 |
| set15 | 1 | 94 | 94.41 | 0.415 | 0.519 |
| set16 | 1 | 148 | 147.7 | 0.566 | 0.452 |

Seasonality of DOC concentration in different depths of Level-1 sites (Hainich, Carlow and Brasschaat) was tested by comparing monthly modelled DOC means versus measurements (Fig. S6).

[Figure]

Figure S6. Monthly DOC means modelled versus measured (mg C L-1) for studied period at a) 20cm of Hainich b)10 to 28cm of Carlow c) 28 to 77 cm of Carlow d) 35cm of Brasschaat e) 75cm of Brasschaat